# Adaptive Memory Retention in Dynamic Graphs

**Fabrizio De Castelli** [1]   **Alessio Gravina** [1]   **Moshe Eliasof** [2,3]   **Carola-Bibiane Schönlieb** [2]   **Davide Bacciu** [1]

## Abstract

Modeling graphs demands a careful balance between long-range propagation of information across nodes and the controlled dissipation of noisy or redundant signals to ensure stable learning and generalization. This challenge is exacerbated in dynamic graphs, where structural and temporal information interact, leading to uncontrolled information accumulation and amplifying noise, thereby affecting generalization. We introduce LAMP, a dynamic graph model for snapshot-based dynamic graphs that incorporates adaptive, learned dissipation within a principled dynamical systems framework. Our architecture combines impulsive neural ODEs with an antisymmetric parameterization to model conservative information flow, alongside data-driven dissipative dynamics that regulate information retention over space and time. This formulation yields stable yet expressive representations and enables effective long-range dependency modeling while avoiding pathological information buildup. We provide a theoretical analysis establishing stability guarantees and characterizing the representational power. Extensive experiments on synthetic and real-world benchmarks demonstrate state-of-the-art performance, particularly on tasks requiring extended-range dependency modeling.

## 1. Introduction

Many real-world systems are inherently dynamic, characterized by data that evolve in both structure and attributes over time (Kazemi et al., 2020; Longa et al., 2023; Gravina & Bacciu, 2024; Jin et al., 2024). Examples include transporta-

[1]Department of Computer Science, University of Pisa, Pisa, Italy [2]Department of Applied Mathematics, University of Cambridge, Cambridge, United Kingdom [3]Faculty of Computer and Information Science, Ben-Gurion University of the Negev, Israel. Correspondence to: Fabrizio De Castelli <fabrizio.decastelli@phd.unipi.it>.

*Proceedings of the 43rd International Conference on Machine Learning*, Seoul, South Korea. PMLR 306, 2026. Copyright 2026 by the author(s).

tion networks with evolving traffic patterns (Li et al., 2018), epidemic spreading over dynamic contact networks (Rozemberczki et al., 2021), and emerging autonomous agentic AI systems operating in complex environments (Kurenkov et al., 2023; Bei et al., 2025). Modeling these systems requires extending static graph learning to dynamic settings. In particular, some approaches disentangle spatial and temporal dynamics using a graph neural network (GNN) and a recurrent neural network (RNN) sequentially (Seo et al., 2018; Li et al., 2018; Zhao et al., 2020; Bai et al., 2021), while others integrate the GNN inside the RNN to jointly capture the temporal evolution and spatial dependencies in the graph (Seo et al., 2018; Chen et al., 2022; Li et al., 2019; Micheli & Tortorella, 2022). Further approaches are extensions of static differential-equation-based GNNs (DE-GNNs) (Han et al., 2023; Hoover et al., 2023) to dynamic graphs (Gravina et al., 2024b; Eliasof et al., 2024b).

Fundamental issues extensively studied in GNNs for static graphs, such as over-smoothing (Cai & Wang, 2020; Rusch et al., 2023), over-squashing (Alon & Yahav, 2021; Di Giovanni et al., 2023a), and vanishing gradients (Arroyo et al., 2025), become far more severe when the temporal dimension comes into play (Marisca et al., 2025). In dynamic graphs, the model must propagate information not only across spatial neighbors but also through long temporal horizons. However, in contrast to the rich literature on effective information propagation in static graphs (Di Giovanni et al., 2023a; Gravina et al., 2023; Shi et al., 2023; Arnaiz-Rodriguez & Errica, 2025; Arroyo et al., 2025), the temporal counterpart has been far less explored. Therefore, addressing this gap is essential, as the inability to effectively propagate information over long horizons directly limits a model's capacity to reason about spatially and temporally distant interactions.

To tackle the challenge of long-range spatio-temporal propagation, we introduce **LAMP** (**L**ong-range **A**daptive **M**emory **P**ropagation), a novel DE-GNN for dynamic graphs with evolving topology and node features, where temporal inputs act as impulses driving the system dynamics. LAMP adaptively balances dissipative and non-dissipative dynamics within its latent state evolution. Specifically, it provides a flexible mechanism to modulate the trade-off between memory retention (conservation) and noise filtering (erasure). We support this design with a theoretical analysis demon-

strating that the model propagates information effectively in the joint spatio-temporal domain by construction.

Our main contributions are as follows: *(i)* We propose LAMP, a novel DE-GNN that integrates non-dissipative dynamics with an adaptive dissipation term. This allows the model to learn the optimal degree of information conservation, seamlessly switching between long-range memory retention and noise filtering. *(ii)* We theoretically prove that LAMP ensures adaptive memory retention that allows the system to selectively filter information. Moreover, LAMP allows for long-range propagation in both space and time. We empirically demonstrate that LAMP preserves stability and keeps hidden states bounded. *(iii)* We introduce a novel set of benchmarks (i.e., 6 tasks) specifically designed to stress-test long-range propagation in both space and time. *(iv)* We conduct extensive experiments to demonstrate that LAMP matches or outperforms state-of-the-art methods across a wide variety of standard temporal graph tasks. Notably, in long-range tasks, LAMP significantly outperforms existing baselines, validating its ability to model distant spatio-temporal interactions.

## 2. LAMP

In this section, we introduce **LAMP** (**L**ong-range **A**daptive **M**emory **P**ropagation), a novel GNN architecture inspired by differential equations, designed to adaptively balance the conservation and erasure of information within its latent state. Figure 1 provides a visual representation of the overall architecture of LAMP.

### 2.1. Notation and Preliminaries

We focus on dynamic graphs with evolving topology and node features, a setting called Discrete-Time Dynamic Graph (D-TDG) (Kazemi et al., 2020; Gravina & Bacciu, 2024) or Snapshot-based Temporal Graphs (Longa et al., 2023). We define a D-TDG as a sequence of graph snapshots $\mathcal{G}_{1:T} = \{\mathcal{G}_1, \ldots, \mathcal{G}_T\}$. Each snapshot $\mathcal{G}_t = (\mathcal{V}, \mathcal{E}(t), \mathbf{X}(t), \mathbf{E}(t))$ consists of a non-empty set of $n$ nodes $\mathcal{V}$, a time-varying edge set $\mathcal{E}(t) \subseteq \mathcal{V} \times \mathcal{V}$, a node feature matrix $\mathbf{X}(t) \in \mathbb{R}^{n \times d_n}$, whose $u$-th row $\mathbf{x}_u(t) \in \mathbb{R}^{d_n}$ represents the input features of node $u$ at time $t$, and an edge feature matrix $\mathbf{E}(t) \in \mathbb{R}^{|\mathcal{E}(t)| \times d_e}$. Additionally, each node $u$ is associated with a hidden state $\mathbf{h}_u(t) \in \mathbb{R}^d$, stacked in a matrix $\mathbf{H}(t)$, which encodes the node state evolution over time $t$. The structural information expressed by $\mathcal{E}(t)$ is encoded in the adjacency matrix $\mathbf{A}(t) \in \{0, 1\}^{n \times n}$, where $[\mathbf{A}(t)]_{uv} = 1$ if $(u, v) \in \mathcal{E}(t)$, and 0 otherwise.

In the following, we use the subscripts $\cdot_{\text{sk}}$ and $\cdot_{\text{sym}}$ to denote antisymmetric and symmetric matrices, respectively.[1]

---

[1]Antisymmetric (i.e., skew-symmetric) matrices satisfy $\mathbf{M}^\top = -\mathbf{M}$, while symmetric matrices satisfy $\mathbf{M}^\top = \mathbf{M}$.

### 2.2. Modeling Graph Snapshots as Impulses

To capture the underlying evolving dynamics of D-TDGs, we combine the perspective of neural ordinary differential equations (Haber & Ruthotto, 2017; Chen et al., 2018) with that of the impulsive ordinary differential equations (I-ODE) (Lakshmikantham et al., 1989). This formulation unifies the internal latent evolution of the graph features with the external arrival of discrete graph snapshots, treating the system as a hybrid dynamical system governed by both continuous flows and discrete events. Therefore, the evolution of $\mathbf{H}(t)$ is governed by a neural impulsive differential equation. For any interval $t \in (t_k, t_{k+1}), k = 1, \ldots, T-1$, the dynamics are conditioned on the graph topology $\mathcal{G}_{t_k}$ observed at the last impulse $t_k$:

$$\begin{cases} \dot{\mathbf{H}}(t) = F_\theta(\mathbf{H}(t); \mathcal{G}_{t_k}) & t \in (t_k, t_{k+1}) \\ \mathbf{H}(t^+) = G_\phi(\mathbf{H}(t^-), \mathbf{X}(t)) & t \in \mathcal{T} \\ \mathbf{H}(0) = \mathbf{H}_0 & t = t_0 \end{cases} \quad (1)$$

where $\mathbf{H}(t^-)$ and $\mathbf{H}(t^+)$ denote the left and right limits of the state trajectory. In this unified framework, $F_\theta : \mathbb{R}^{n \times d} \to \mathbb{R}^{n \times d}$ is a neural vector field parameterized by the set $\theta$ that serves as spatial diffusive operator, while $G_\phi : \mathbb{R}^{n \times d} \times \mathbb{R}^{n \times d_n} \to \mathbb{R}^{n \times d}$ is the impulse operator, a neural network with set of parameters $\phi$ responsible for incorporating the new observed features in the hidden state.

Specifically, we define LAMP through a first-order neural I-ODE that enables temporal diffusion at a pace dictated by the sequence, and spatial propagation obtained by solving $\dot{\mathbf{H}}(t) = F_\theta(\mathbf{H}(t); \mathcal{G}_{t_k})$ in Equation (1) for each new incoming snapshot in the interval between two graph snapshots, $t \in [t_k, t_{k+1}]$.

To effectively propagate information jointly in space and time, we leverage the theory of non-dissipative neural ODEs (Haber & Ruthotto, 2017; Chang et al., 2019; Gravina et al., 2023; 2025a), which links non-dissipative behavior to the stability of the system and to the sensitivity of its solutions to initial conditions. This sensitivity is governed by the Jacobian eigenvalues, with purely imaginary spectra ensuring stable dynamics that preserve information throughout the evolution. Accordingly, we define $F_\theta$ as the following graph neural ODE, with parameters $\theta = \{\mathbf{W}_{\text{sk}}, \mathbf{V}_{\text{sk}}, \mathbf{Z}_{\text{sym}}, \boldsymbol{\Gamma}\}$:

$$\begin{aligned} F_\theta(\mathbf{H}(t); \mathcal{G}_{t_k}) = \sigma\Big( & \mathbf{H}(t)\mathbf{W}_{\text{sk}} \\ & + \mathcal{S}(\mathbf{A}(t_k), \mathbf{H}(t); \mathbf{V}_{\text{sk}}) \quad (2) \\ & + \mathcal{K}(\mathbf{A}(t_k), \mathbf{H}(t); \mathbf{Z}_{\text{sym}})\Big), \end{aligned}$$

with $\sigma$ the activation function. $\mathcal{S}$ and $\mathcal{K}$ are permutation-invariant neighborhood aggregation functions, which define the local message-passing dynamics over the graph. While $\mathcal{S}$ and $\mathcal{K}$ refer to generic aggregation functions, in this paper

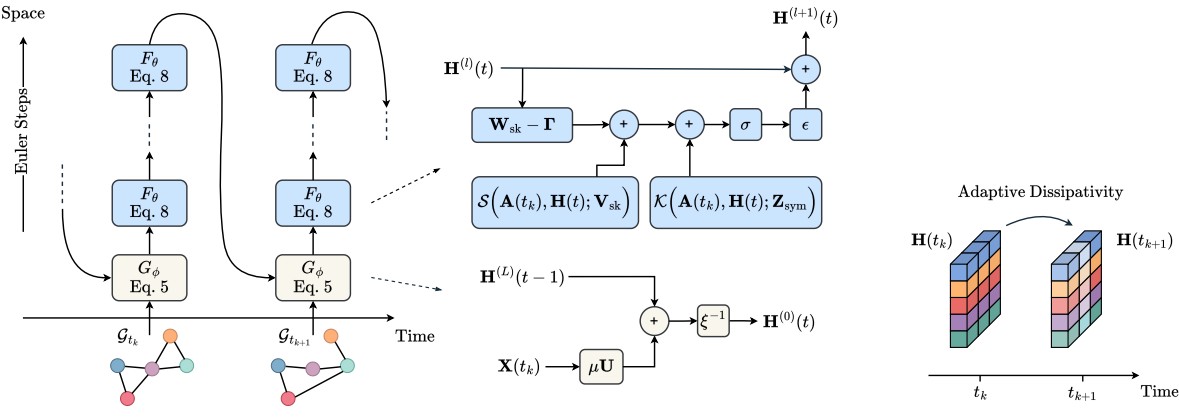

*(a)* High-level architecture.

*(b)* Adaptive channel-wise dissipativity.

*Figure 1.* Overall architecture of LAMP. (a) On the left, a high-level block architecture showing components across space and time. On the top-right, visual explanation of the continuous flow $F_\theta$ of Equation (8). Finally, on the bottom-right, representation of the impulse operator $G_\phi$ of Equation (5) at impulse timesteps. (b) The effect of adaptive dissipativity in LAMP, showing latent hidden state channels fading between times $t_k$ and $t_{k+1}$.

we explore the following parameterizations that ensure that the Jacobian of Equation (2) has purely imaginary eigenvalues and the system possesses a non-dissipative dynamics:

$$\mathcal{S}(\mathbf{A}, \mathbf{H}; \mathbf{V}_{\text{sk}}) = -\mathbf{A}_{\text{sym}}\mathbf{H}\mathbf{V}_{\text{sk}}, \tag{3}$$

$$\mathcal{K}(\mathbf{A}, \mathbf{H}; \mathbf{Z}_{\text{sym}}) = \beta\left(\mathbf{A}_{\text{rw}} - (\mathbf{A}_{\text{rw}})^\top\right)\mathbf{H}\mathbf{Z}_{\text{sym}}, \tag{4}$$

where $\mathbf{A}_{\text{sym}} = \mathbf{D}^{-\frac{1}{2}}\mathbf{A}\mathbf{D}^{-\frac{1}{2}}$ is the symmetrically normalized adjacency, $\mathbf{A}_{\text{rw}} = \mathbf{D}^{-1}\mathbf{A}$ is the random-walk normalized adjacency, and $\beta \in \mathbb{R}$ is a hyperparameter that controls the magnitude of $\mathcal{K}$.[2] We note that such formulation ensures purely imaginary spectra of the Jacobian. For an in-depth analysis of the Jacobian of our LAMP, we refer the reader to Appendix A.1.

At observation time $t_{k+1}$, the continuous evolution is interrupted by the instantaneous jump map $G_\phi$, which integrates newly arriving external information into the latent state. Specifically, this operator fuses the latent state available just before the new input $\mathbf{H}(t_k^-)$ with the new observed node features $\mathbf{X}(t_k)$. We implement this update as a linear injection with parameters $\phi = \mathbf{U}$:

$$G_\phi(\mathbf{H}(t_k^-), \mathbf{X}(t_k)) = \frac{1}{\xi}\left(\mathbf{H}(t_k^-) + \mu\mathbf{X}(t_k)\mathbf{U}\right), \tag{5}$$

where the scalars $\xi, \mu \in \mathbb{R}$ act as normalization hyperparameters. The impulse operator is visually represented at the bottom of Figure 1. We discuss their role and theoretical implications in Section 2.3.

[2] We observe that temporal edge features can be easily incorporated into Equations (3) and (4), since scalar features can be integrated in the adjacency matrix, while multivariate features can be projected to $d$-dimensions and summed during convolution, as with classical MPNNs (Gilmer et al., 2017; Hu et al., 2020).

**Adaptive Dissipativity.** To enable adaptive control over information preservation and decay, we relax the conservative channel mixing in Equation (2) (i.e., $\mathbf{H}(t)\mathbf{W}_{\text{sk}}$) by subtracting a learned diagonal matrix

$$\mathbf{\Gamma} = \text{diag}(\text{sigmoid}(\gamma)) \tag{6}$$

with $\gamma \in \mathbb{R}^d$. Therefore, Equation (2) can be rewritten as

$$\begin{aligned} F_\theta(\mathbf{H}(t); \mathcal{G}_{t_k}) = \sigma\Big( & \mathbf{H}(t)(\mathbf{W}_{\text{sk}} - \mathbf{\Gamma}) \\ & + \mathcal{S}(\mathbf{A}(t_k), \mathbf{H}(t); \mathbf{V}_{\text{sk}}) \\ & + \mathcal{K}(\mathbf{A}(t_k), \mathbf{H}(t); \mathbf{Z}_{\text{sym}})\Big). \end{aligned} \tag{7}$$

This mechanism (summarized visually in Figure 1) allows the model to continuously and dimension-wise balance dissipative and non-dissipative dynamics during latent-state evolution, effectively regulating how much information is preserved or attenuated as it propagates through time and across the graph. To control the dissipation rate at the beginning of training, we initialize $\gamma$ as the inverse of the sigmoid function. A theoretical discussion on the impact of the adaptive dissipative mechanism is provided in Section 2.3.

**Numerical Discretization and Implementation.** To transition from the continuous I-ODE formulation to discrete latent state updates, we discretize the evolution over the interval $\Delta t = t_{k+1} - t_k$ using the forward Euler method (Ascher, 2008) with a fixed step size $\epsilon$. We perform $L$ discretization steps over the interval $[t, t + \Delta t]$, thus assuming $\Delta t \approx \epsilon L$.

Denoting the discretized hidden state at step (layer) $l$ and time $t$ as $\mathbf{H}^{(l)}(t)$, the Euler update rule reads:

$$\mathbf{H}^{(l+1)}(t) = \mathbf{H}^{(l)}(t) + \epsilon F_\theta(\mathbf{H}^{(l)}(t); \mathcal{G}_t). \tag{8}$$

The boundary condition $\mathbf{H}^{(0)}(t)$ is given by Equation (5), which combines the current node input features $\mathbf{X}(t)$ with the latent node representations propagated from the previous snapshot at $t - \Delta t$. This ensures that the node evolution over the interval $[t, t + \Delta t]$ is initialized with both past memory and newly observed information. We remark that all weights in $F_\theta$ and $G_\phi$ are shared across discretization and time steps. An illustration of the interplay between $F_\theta$ and $G_\phi$ is provided in Figure 1.

Lastly, we note that our definition in Equation (7) ensures stability of the forward Euler method[3] by construction, as the eigenvalues of the Jacobian are imaginary but shifted on the stability region proportionally to the diagonal matrix $\mathbf{\Gamma}$ (see Appendix A.1.2).

## 2.3. Theoretical properties of LAMP

We now provide theoretical statements about information conservation, dissipation, and stability of LAMP, showing that our model effectively performs long-range propagation between nodes while controlling the amount of information to conserve in each channel.

**Adaptive Memory Retention.** While strictly non-dissipative propagation preserves the system information indefinitely, a property advantageous in deep learning for both time series and static graphs (Haber & Ruthotto, 2017; Chang et al., 2019; Gravina et al., 2023; 2025a; Hariri et al., 2025), it poses distinct challenges in temporal settings. In sequence learning, strictly conservative dynamics retain all historical interactions with equal magnitude, leading to an unbounded accumulation of noise that obscures recent signals (Pascanu et al., 2013). Similarly, it has been shown that contractivity can help in graph learning (Eliasof et al., 2024c). Conversely, uncontrolled dissipation causes premature memory collapse, preventing the capture of long-range dependencies (Hochreiter & Schmidhuber, 1997), that is also apparent in graph learning (Eliasof et al., 2023; Di Giovanni et al., 2023b).

LAMP addresses this trade-off via the learnable damping term $\mathbf{\Gamma}$ in Equation (6), which allows the system to selectively filter information. To formalize this behavior, we utilize contraction theory (Lohmiller & Slotine, 1998). A nonlinear system $\dot{\mathbf{h}} = f(\mathbf{h})$ is contracting if all trajectories converge towards each other exponentially fast, a property guaranteed if the logarithmic norm of the Jacobian, $\mu_2(\mathbf{J}_{F_\theta})$, is uniformly negative. To apply this to LAMP, we first start by providing the exact definition of the continuous flow Jacobian at a fixed time $t$. Let $\mathbf{h} = \text{vec}(\mathbf{H}(t))$ be the vectorized

state. The Jacobian of the vector field is given by

$$\mathbf{J}_{F_\theta}(\mathbf{h}) = \mathbf{D}(\mathbf{h})(\mathbf{M}_{\text{sk}} - \mathbf{\Gamma} \otimes \mathbf{I}_n), \qquad (9)$$

where $\mathbf{D}(\mathbf{h}) = \text{diag}(\sigma'(\mathbf{h}))$ is the matrix of activation derivatives, and $\mathbf{M}_{\text{sk}}$ is the total skew-symmetric mixing, defined as

$$\begin{aligned} \mathbf{M}_{\text{sk}} = \left(\mathbf{W}_{\text{sk}}^\top \otimes \mathbf{I}_n\right) + \left(\mathbf{V}_{\text{sk}}^\top \otimes \mathbf{A}_{\text{sym}}\right) \\ + \beta\left(\mathbf{Z}_{\text{sym}}^\top \otimes \left(\mathbf{A}_{\text{rw}} - \mathbf{A}_{\text{rw}}^\top\right)\right) \end{aligned} \qquad (10)$$

with $\otimes$ denoting the Kronecker product. The exact derivation and spectral properties of the Jacobian are left to Appendix A.1. Now we bound the contraction rate of LAMP by analyzing the competition between the instantaneous expansion induced by the state-dependent activations and the effective dissipation provided by the learned damping.

**Theorem 2.1** (Adaptive Memory Retention). *Let $\mathbf{G} = \mathbf{\Gamma} \otimes \mathbf{I}_n$. The instantaneous rate of change of LAMP, $\mu_2(\mathbf{J}_{F_\theta})$, is strictly bounded by:*

$$\mu_2(\mathbf{J}_{F_\theta}) \leq \nu(\mathbf{h}) - \min_i \left([\mathbf{D}(\mathbf{h})]_{ii}[\mathbf{G}]_{ii}\right), \qquad (11)$$

*where the coupling expansion rate $\nu(\mathbf{h})$ is defined by the symmetric interaction between the activation derivatives and the mixing matrix:*

$$\nu(\mathbf{h}) = \lambda_{\max}\left(\frac{\mathbf{D}(\mathbf{h})\mathbf{M}_{sk} - \mathbf{M}_{sk}\mathbf{D}(\mathbf{h})}{2}\right). \qquad (12)$$

*Consequently, strict contraction ($\mu_2 < 0$) is guaranteed whenever the dissipation is stronger than the expansion rate.*

The proof is in Appendix A.2. This result highlights the dual role of the damping term $\mathbf{\Gamma}$. First, it provides a theoretical guarantee of stability: by strictly bounding the Jacobian's spectrum in the negative half-plane, $\mathbf{\Gamma}$ prevents the pathological signal explosion often seen in conservative systems. Second, and crucially, it allows the model to learn a task-driven *timescale* of information retention. The term $\min_i \left([\mathbf{D}(\mathbf{h})]_{ii}[\mathbf{G}]_{ii}\right)$ effectively sets a lower bound on the forgetting rate. When this term sufficiently counterbalances the expansion rate, LAMP shifts the system dynamics arbitrarily close to the imaginary axis, thereby controlling the preservation of long-range dependencies while retaining the ability to locally increase damping to suppress noise. In Appendix B.1 we provide empirical evidence that this property holds.

**Boundedness and Stability.** To strengthen our analysis, we now derive the conditions required for a globally stable discrete process. While an optimal balance between memory retention and erasure can be achieved by tuning $\mathbf{\Gamma}$ arbitrarily close to the coupling expansion rate, the previous analysis was restricted to the continuous integration interval. Since

---

[3]The forward Euler method is stable if the spectra of the system's Jacobian, scaled by the step size, reside within the unit circle centered at $(-1, 0)$ (Ascher & Petzold, 1998).

our system is open and constantly receives information injections from new graph snapshots $\mathbf{X}(t_k)$, we must establish a global boundedness property. To guarantee that the latent state does not diverge over infinitely long sequences, we leverage the framework of Input-to-State Stability (ISS) (Sontag, 1989). ISS extends classical stability to open systems, requiring that the state magnitude remains bounded by a function of the input magnitude (the asymptotic gain) plus a decaying transient term.

Because LAMP processes inputs via discrete steps, we analyze the asymptotic gain property for discrete-time nonlinear systems (Jiang & Wang, 2001). We derive the precise condition under which this property is guaranteed.

**Proposition 2.2** (Global Boundedness). *Assume bounded inputs such that* $\sup_t \|\mathbf{X}(t)\|_F \leq B$ *and a fixed sampling interval* $\Delta t \approx \epsilon L$. *Let* $\omega = \sup_{\mathbf{h}} \mu_2(\mathbf{J}_{F_\theta})$ *be the maximum logarithmic norm of the vector field over the sequence. If the memory normalization satisfies the stability condition* $\xi > e^{\omega \epsilon L}$, *the latent state* $\mathbf{H}(t)$ *remains globally bounded as* $t \to \infty$, *satisfying:*

$$\limsup_{t \to \infty} \|\mathbf{H}(t)\|_F \leq \frac{\mu \|\mathbf{U}\|_F B}{\xi - e^{\omega \epsilon L}}. \qquad (13)$$

The proof is in Appendix A.3. This inequality represents the discrete asymptotic gain of the system, quantifying the trade-off between stability and memory capacity. The numerator represents the input energy injection, scaled by the encoder amplification $\mu \|\mathbf{U}\|_F$. The denominator, $\xi - e^{\omega \epsilon L}$, defines the effective boundedness margin. Thus, the system's behavior is governed by the interplay between the memory normalization $\xi$ and the maximum expansion rate $\omega$. Whenever $\xi > e^{\omega \epsilon L}$, LAMP guarantees boundedness over infinite sequences. Specifically, when capturing long-range temporal dependencies, the model can locally learn the optimal erasure rate during spatial diffusion, preserving signal magnitude across impulses without risking divergence. Since the memory normalization condition explicitly depends on learned weights, we empirically demonstrate in Appendix B.2 that our model maintains bounded trajectories in practice, without requiring weight normalization, gradient clipping (Pascanu et al., 2013), or scheduling of any term in the bound.

**Sensitivity Analysis.** Over-squashing, and, more generally, effective propagation between nodes, has been studied in static GNNs through sensitivity analysis (Topping et al., 2022; Di Giovanni et al., 2023a), as well as for spatio-temporal GNNs where only node features evolve in time (Marisca et al., 2025). In the static case, this analysis shows that classical GNNs typically exhibit an exponentially decaying information-propagation rate (Gravina et al., 2025a), leading distant nodes to become progressively less effective at sharing new information. To quantify sensitivity for

our LAMP, we are interested in exploring how previously computed representations affect information propagation across time and space. In particular, we measure how the hidden representation of a node $v$ at time $t$ can be affected by the representation of another node $u$ at an arbitrary time $s < t$, such that there exists a time-respecting path between $u$ and $v$. We incorporate this analysis into our model, which can process D-TDGs in which both the topology and node features evolve over time. In the following theorem, we present the sensitivity bound for LAMP, indicating effective propagation between nodes.

**Theorem 2.3** (Sensitivity Bound). *Let* $[s, t] \subseteq [t_1, t_T]$ *be an arbitrary sub-interval of the input sequence times and let* $s \leq t_i, \ldots, t_j \leq t \in \mathcal{T}$ *be the discrete times at which impulses occur in the interval. Then, the LAMP sensitivity of node* $v$ *with respect to node* $u$ *is bounded by:*

$$\left\| \frac{\partial \mathbf{h}_v(t)}{\partial \mathbf{h}_u(s)} \right\|_2 \leq \underbrace{\xi^{-(j-i+1)}(c_\sigma w d)^{t-s}}_{model} \underbrace{\mathbf{\Psi}_{uv}}_{topology} . \qquad (14)$$

*Here,* $c_\sigma$ *is the Lipschitz constant of the activation* $\sigma$ *and* $w \in \mathbb{R}$ *is the maximal entry-value over all weight matrices. The matrix* $\mathbf{\Psi} \in \mathbb{R}^{n \times n}$ *is the chain multiplication of each message passing matrix adopted at impulse times:*

$$\mathbf{\Psi} = \mathbf{S}_{j+1}^{t-t_j} \left[ \prod_{k=j}^{i+1} \mathbf{S}_k^{\Delta t} \right] \mathbf{S}_i^{t_i - s},$$

$$\mathbf{S}_k = \mathbf{I} + \mathbf{A}_{sym}(t_k) + \beta \left( \mathbf{A}_{rw}(t_k) - (\mathbf{A}_{rw}(t_k))^\top \right).$$

We leave the proof to Appendix A.4. Importantly, in Equation (14) the maximal weight value $w$ incorporates the dissipation term $\mathbf{\Gamma}$. This suggests that the model's sensitivity is not limited by a rigid, pre-defined decay. The architecture allows for lossless propagation by default, restricting information flow only when the dissipation weights are learned to be the dominant factor in the system. Unlike standard recurrent architectures where information loss is often inherent to stability constraints, LAMP effectively decouples stability from memory capacity. The information-retention potential is determined solely by the topology $\mathbf{\Psi}$ and the sequence length, implying that the underlying dynamics can preserve signals indefinitely. In this framework, the learned dissipation $\mathbf{\Gamma}$ (analyzed in Theorem 2.1) acts not as a structural bottleneck, but as an adaptive filter: it selectively removes specific signals deemed noisy or redundant, while leaving the intrinsic long-range propagation paths intact.

**Complexity Analysis.** The computational complexity of LAMP is determined by the numerical integration of the vector field $F_\theta$. A single evaluation of Equation (8) exhibits runtime and memory consumption comparable to a standard MPNN (Gilmer et al., 2017). Consequently, it scales linearly with the number of nodes ($n$) and edges

($|\mathcal{E}|$), yielding to a cost of $\mathcal{O}(|\mathcal{E}|d + nd^2)$, where $d$ is the latent dimension. Performing $L$ Euler steps per input snapshot across $T$ timestamps, the total time complexity results $\mathcal{O}(T(L(|\mathcal{E}|d + nd^2) + m))$ with $m$ the complexity of Equation (5). Therefore, LAMP's complexity remains strictly below that of Transformer-based spatio-temporal models (Gao et al., 2024; Jiang et al., 2023; Liu et al., 2023) since it does not introduce costly global operations (i.e., attention over all node pairs). Concerning memory consumption, we note that LAMP has a total complexity of $\mathcal{O}(3d^2 + d + d \cdot d_n)$, arising from the three hidden-state transformations in Equation (7), the adaptive dissipative term in Equation (6), and the input transformation in Equation (5). Lastly, we observe that since the three hidden-state transformations involve symmetric or antisymmetric matrices, only half of each matrix can be stored, further reducing the memory footprint. We report the empirical evaluation of LAMP's runtimes in Appendix B.3.

## 3. Related Work

**Long-Range Propagation in Graphs.** While GNNs effectively model local graph structures via message passing, their localized nature may hinder capturing interactions between distant nodes (Shi et al., 2023; Arroyo et al., 2025; Miglior et al., 2026). In static graphs, this limitation often manifests as over-squashing (Alon & Yahav, 2021; Topping et al., 2022; Di Giovanni et al., 2023a) and over-smoothing (Cai & Wang, 2020; Oono & Suzuki, 2020; Rusch et al., 2023), both of which are closely linked to the vanishing gradient problem (Arroyo et al., 2025). To mitigate these issues, a wide range of approaches has been proposed, including graph rewiring techniques (Gasteiger et al., 2019; Topping et al., 2022; Karhadkar et al., 2023; Barbero et al., 2024) and Graph Transformers (Rampášek et al., 2022; Shi et al., 2021; Shirzad et al., 2023; Ying et al., 2021) which alter the graph topology to facilitate communication between nodes either by rewiring or attention mechanisms. Alternative methods propose filtering messages during information flow (Errica et al., 2024; Finkelshtein et al., 2024), designing stable architectures (Gravina et al., 2023; 2025b;a; Leeney et al., 2025), learning adaptive graph filtering operators (Eliasof et al., 2025), exploiting port-Hamiltonian dynamics (Heilig et al., 2025; Trenta et al., 2025; Hoang et al., 2026), or introducing virtual nodes to ease long-range communication (Southern et al., 2025). However, these approaches are restricted to static graphs. When graphs evolve, these challenges are further exacerbated. Even in the simplified setting of dynamic graphs with time-varying node features only, recent work has shown that the temporal dimension introduces additional information propagation issues, amplifying the dissipative effects already observed in static graphs (Marisca et al., 2025). To address these complex spatio-temporal dependencies, recent works have proposed extending positional and

structural encodings to the time domain, such as adapting Laplacian Positional Encodings to capture temporal evolution (Galron et al., 2026), leveraging purely conservative dynamics to learn long-range dependencies in continuous-time dynamic graphs (C-TDGs) (Gravina et al., 2024a), or exploiting transition graphs to favor long-range information propagation (Zheng et al., 2024). In contrast, LAMP achieves long-range propagation in D-TDGs with evolving topology and features by adaptively regulating information preservation and decay, while remaining scalable with respect to the number of nodes and edges.

**Adaptive Dissipativity.** Although dissipativity is a foundational concept in control theory for analyzing system stability (Willems, 1972), its explicit integration into deep learning architectures remains under-explored. In the realm of RNNs and GNNs, stability is often sought via contractivity analysis (Miller & Hardt, 2019; Revay & Manchester, 2020), which guarantees that input or state perturbations vanish over time, thereby preventing gradient explosion. However, strict contraction can hinder effective learning and is directly correlated with vanishing gradients (Bengio et al., 1994; Pascanu et al., 2013). Existing methods typically navigate this trade-off through static architectural constraints, such as orthogonal (Arjovsky et al., 2016; Vorontsov et al., 2017; Helfrich et al., 2018) or antisymmetric (Chang et al., 2019) parameterizations. Similarly, sequence models like in (Feldman et al., 2025) seek to balance this trade-off by adaptively sampling historical data to filter interactions, to modulate the information flow. While adaptive diffusion rates have been explored in static graph learning (Choi et al., 2023; Eliasof et al., 2023; 2024a), to the best of our knowledge, no prior work has investigated a mechanism capable of dynamically controlling the degree of dissipation within DE-GNNs for dynamic graph learning.

## 4. Experiments

In this section, we discuss the empirical assessment of LAMP. In Section 4.1, we evaluate our method on a novel synthetic benchmark called `ColoredLeafCounting` specifically designed to stress-test the effectiveness of models in long-range propagation over space and time, as well as reporting an ablation on the importance of the adaptive dissipativity. To further show the effectiveness of LAMP, we evaluate it on popular real-world D-TDG forecasting benchmarks in Section 4.2.1, and link prediction tasks in Section 4.2.2. An additional ablation regarding runtimes is reported in Appendix B.3, while regarding step size, number of discretization steps, and additional hyperparameters in Appendix B.5. Details regarding baselines, dataset statistics, and hyperparameter grid search configurations can be found in Appendix C. All experiments were performed on a server with NVIDIA V100 GPUs. We

openly release the code to reproduce our experiments at `https://github.com/FabriDeCastelli/LAMP`. The `ColoredLeafCounting` benchmark is also available at `https://huggingface.co/datasets/FabrizioDeCastelli/ColoredLeafCounting`.

### 4.1. Colored Leaf Counting Long-Range Benchmark

To rigorously evaluate long-range propagation jointly in time and space, we introduce a novel synthetic benchmark: `ColoredLeafCounting`. Specifically, the benchmark considers a rooted tree where the root node and all leaf nodes are colored, while internal nodes act solely as intermediaries for message passing. Over a sequence of discrete timesteps, each leaf node may emit a signal (i.e., a "blink"), independently across time. At the end of the temporal sequence, the objective is to predict the total number of blinks produced over time by leaf nodes whose color matches that of the root. Solving this task requires the model to propagate information across long spatial paths (from leaves to the root) while simultaneously aggregating signals over multiple timesteps, making it a challenging testbed for joint spatio-temporal propagation.

We consider three different sequence lengths $T \in \{200, 400, 600\}$ and two sets of node colors, i.e., the binary color setting {black, white} or the three color setting {red, green, blue}. Therefore, `ColoredLeafCounting` consists of a total of 6 different tasks with increased complexity. Additional details about `ColoredLeafCounting` are reported in Appendix C.4.

**Results.** In Tables 1 and 2, we report the mean and standard deviation of the test MSE. Remarkably, our model significantly outperforms baselines across all configurations.

In the binary color setting (Table 1), which reduces to global signal counting since the root node should count all blinks in the timeseries, LAMP achieves the best performance. Notably, baselines exhibit performance degradation as $T$ increases, indicating difficulty propagating leaf signals to the root over long sequences. When task complexity increases (Table 2), i.e., in the multi-color setting, the model must implicitly filter signals from mismatched colors. As shown in the results, baselines fail to filter this noise effectively, leading to high error rates. In contrast, LAMP maintains low error even at $T = 600$. This validates the importance of the adaptive dissipation mechanism, allowing the LAMP to selectively filter noise (wrong colors) while the conservative dynamics preserve the target signal over long horizons.

**Ablation on the Adaptive Dissipativity.** We analyze the role of the dissipation mechanism by progressively relaxing LAMP into five variants: (i) the full model with adaptive dissipation (LAMP$_{\text{adaptive}}$, learned $\mathbf{\Gamma}$); (ii) fixed dissipation (LAMP$_{\text{fixed}}$, $\mathbf{\Gamma}$ fixed); (iii) no dissipation (LAMP$_{\mathbf{\Gamma}=0}$,

*Table 1.* Test MSE ($\downarrow$) on the `ColoredLeafCounting` benchmark for the binary color setting, averaged over 5 runs. **First**, **second**, and **third** results are color-coded.

| Model | $T = 200$ | $T = 400$ | $T = 600$ |
|---|---|---|---|
| A3T-GCN | $\mathbf{0.792}_{\pm 0.002}$ | $0.774_{\pm 0.018}$ | $1.374_{\pm 0.001}$ |
| DCRNN | $0.817_{\pm 0.038}$ | $0.779_{\pm 0.025}$ | $1.373_{\pm 0.003}$ |
| DynGESN | $0.806_{\pm 0.009}$ | $0.771_{\pm 0.004}$ | $1.373_{\pm 0.000}$ |
| EvolveGCN-H | $0.806_{\pm 0.037}$ | $0.767_{\pm 0.013}$ | $1.375_{\pm 0.001}$ |
| EvolveGCN-O | $0.801_{\pm 0.028}$ | $0.767_{\pm 0.015}$ | $1.375_{\pm 0.001}$ |
| GC-LSTM | $\mathbf{0.791}_{\pm 0.019}$ | $0.768_{\pm 0.017}$ | $1.374_{\pm 0.001}$ |
| GCRN-GRU | $0.806_{\pm 0.017}$ | $\mathbf{0.765}_{\pm 0.007}$ | $1.374_{\pm 0.003}$ |
| GCRN-LSTM | $0.799_{\pm 0.011}$ | $0.770_{\pm 0.012}$ | $1.376_{\pm 0.004}$ |
| T-GCN | $\mathbf{0.792}_{\pm 0.002}$ | $0.764_{\pm 0.001}$ | $1.374_{\pm 0.002}$ |
| LAMP | $0.467_{\pm 0.083}$ | $0.394_{\pm 0.130}$ | $0.831_{\pm 0.136}$ |

*Table 2.* Test MSE ($\downarrow$) on the `ColoredLeafCounting` benchmark for the three-color setting, averaged over 5 runs. **First**, **second**, and **third** results are color-coded.

| Model | $T = 200$ | $T = 400$ | $T = 600$ |
|---|---|---|---|
| A3T-GCN | $1.304_{\pm 0.009}$ | $1.003_{\pm 0.002}$ | $0.836_{\pm 0.003}$ |
| DCRNN | $\mathbf{1.283}_{\pm 0.023}$ | $0.811_{\pm 0.177}$ | $0.675_{\pm 0.043}$ |
| DynGESN | $1.303_{\pm 0.008}$ | $1.010_{\pm 0.008}$ | $0.838_{\pm 0.002}$ |
| EvolveGCN-H | $1.305_{\pm 0.017}$ | $\mathbf{0.991}_{\pm 0.000}$ | $0.868_{\pm 0.001}$ |
| EvolveGCN-O | $1.307_{\pm 0.018}$ | $\mathbf{0.991}_{\pm 0.000}$ | $0.868_{\pm 0.001}$ |
| GC-LSTM | $1.293_{\pm 0.020}$ | $1.004_{\pm 0.013}$ | $0.837_{\pm 0.004}$ |
| GCRN-GRU | $1.303_{\pm 0.003}$ | $1.003_{\pm 0.010}$ | $\mathbf{0.830}_{\pm 0.017}$ |
| GCRN-LSTM | $1.248_{\pm 0.078}$ | $1.012_{\pm 0.008}$ | $0.837_{\pm 0.002}$ |
| T-GCN | $1.307_{\pm 0.010}$ | $1.004_{\pm 0.005}$ | $0.838_{\pm 0.002}$ |
| LAMP | $0.057_{\pm 0.015}$ | $0.264_{\pm 0.199}$ | $0.223_{\pm 0.071}$ |

$\mathbf{\Gamma} = 0$); (iv) no dissipation and no weight constraints on $\mathbf{W}_{\text{sk}}, \mathbf{V}_{\text{sk}}, \mathbf{Z}_{\text{sym}}$ (LAMP$_{\text{NoWC}}$); and (v) no dissipation, no weight constraints, and no antisymmetric aggregation (LAMP$_{\text{NoWC},\beta=0}$). Table 3 reports the mean and standard deviation of the test MSE across these configurations. The results isolate the contribution of each theoretical component of LAMP. Removing both the weight constraints and the antisymmetric aggregation produces the largest degradation, confirming that the non-dissipative parameterization is the dominant driver of long-range propagation. Without the non-dissipative parametrization, the model fails to retain leaf signals across long horizons. Among the top three configurations (adaptive, fixed, $\mathbf{\Gamma} = 0$), the differences remain within standard deviation margins on most settings. Notably, the fully non-dissipative variant ($\mathbf{\Gamma} = 0$) performs favorably in some tasks, as they inherently favor information retention over dissipation.

### 4.2. Real-World Benchmarks

We evaluate LAMP on real-world benchmarks spanning both node-level and link-level tasks. In the former, the objective is to forecast future node values, while in the latter, the goal is to predict the time-evolving graph topology.

*Table 3.* Test MSE (↓) on `ColoredLeafCounting` (5 runs). Best scores in **green**. Variants progressively relax the model: *adaptive* (full LAMP), *fixed* ($\Gamma$ fixed), $\Gamma = 0$ (non-dissipative), *NoWC* (no weight constraints, $\Gamma = 0$), and *NoWC*, $\beta = 0$ (no weight constraints, $\Gamma = 0$, and no antisymmetric aggregation).

| Model | $T = 200$ | $T = 400$ | $T = 600$ |
|---|---|---|---|
| **Three-colors** | | | |
| LAMP$_{adaptive}$ | $0.057_{\pm 0.015}$ | $\mathbf{0.264}_{\pm 0.199}$ | $0.223_{\pm 0.071}$ |
| LAMP$_{fixed}$ | $0.065_{\pm 0.030}$ | $0.523_{\pm 0.291}$ | $0.235_{\pm 0.080}$ |
| LAMP$_{\Gamma=0}$ | $\mathbf{0.051}_{\pm 0.008}$ | $0.520_{\pm 0.145}$ | $\mathbf{0.212}_{\pm 0.064}$ |
| LAMP$_{NoWC}$ | $1.327_{\pm 0.020}$ | $1.071_{\pm 0.017}$ | $0.650_{\pm 0.257}$ |
| LAMP$_{NoWC,\beta=0}$ | $1.330_{\pm 0.021}$ | $1.053_{\pm 0.026}$ | $0.718_{\pm 0.178}$ |
| **Binary-color** | | | |
| LAMP$_{adaptive}$ | $0.467_{\pm 0.083}$ | $\mathbf{0.394}_{\pm 0.130}$ | $\mathbf{0.831}_{\pm 0.136}$ |
| LAMP$_{fixed}$ | $\mathbf{0.421}_{\pm 0.063}$ | $0.432_{\pm 0.105}$ | $1.135_{\pm 0.239}$ |
| LAMP$_{\Gamma=0}$ | $0.438_{\pm 0.054}$ | $0.438_{\pm 0.091}$ | $0.983_{\pm 0.422}$ |
| LAMP$_{NoWC}$ | $0.848_{\pm 0.017}$ | $0.797_{\pm 0.003}$ | $1.373_{\pm 0.000}$ |
| LAMP$_{NoWC,\beta=0}$ | $0.847_{\pm 0.018}$ | $0.793_{\pm 0.009}$ | $1.373_{\pm 0.001}$ |

*Table 4.* Test MSE (↓) on Wikipedia Math, ChickenPox Hungary, and Twitter Tennis, averaged over 10 runs. **First**, **second**, and **third** best results are color-coded. Baseline results are reported from (Micheli & Tortorella, 2022; Errica et al., 2023).

| Model | Wiki Math | Tennis | Chickenpox |
|---|---|---|---|
| Mean baseline | 0.843 | 0.482 | 1.117 |
| Linear baseline | 0.663 | 0.356 | 0.952 |
| DCRNN | $0.679_{\pm 0.007}$ | $0.478_{\pm 0.004}$ | $1.097_{\pm 0.006}$ |
| GCRN-GRU | $0.680_{\pm 0.021}$ | $0.477_{\pm 0.007}$ | $1.103_{\pm 0.004}$ |
| GCRN-LSTM | $0.678_{\pm 0.008}$ | $0.477_{\pm 0.006}$ | $1.097_{\pm 0.006}$ |
| GC-LSTM | $0.677_{\pm 0.009}$ | $0.475_{\pm 0.010}$ | $1.095_{\pm 0.005}$ |
| DyGrAE | $0.621_{\pm 0.012}$ | $0.480_{\pm 0.005}$ | $1.102_{\pm 0.013}$ |
| EvolveGCN-H | $0.779_{\pm 0.031}$ | $0.481_{\pm 0.003}$ | $1.137_{\pm 0.026}$ |
| EvolveGCN-O | $0.807_{\pm 0.047}$ | $0.484_{\pm 0.002}$ | $1.135_{\pm 0.011}$ |
| A3T-GCN | $0.618_{\pm 0.008}$ | $0.477_{\pm 0.005}$ | $1.078_{\pm 0.009}$ |
| T-GCN | $0.616_{\pm 0.011}$ | $0.478_{\pm 0.004}$ | $1.083_{\pm 0.011}$ |
| MPNN LSTM | $0.856_{\pm 0.021}$ | $0.482_{\pm 0.001}$ | $1.125_{\pm 0.005}$ |
| DynGESN | $\mathbf{0.610}_{\pm 0.003}$ | $\mathbf{0.300}_{\pm 0.003}$ | $\mathbf{0.907}_{\pm 0.007}$ |
| HMM4G | $\mathbf{0.542}_{\pm 0.008}$ | $\mathbf{0.333}_{\pm 0.004}$ | $\mathbf{0.939}_{\pm 0.013}$ |
| LAMP | $\mathbf{0.444}_{\pm 0.003}$ | $\mathbf{0.304}_{\pm 0.001}$ | $\mathbf{0.748}_{\pm 0.006}$ |

### 4.2.1. NODE-LEVEL PREDICTION

For the node-level prediction tasks, we consider five popular benchmarks, i.e., Wikipedia Math, ChickenPox Hungary, and Twitter Tennis (Rozemberczki et al., 2021), as well as Metr-LA and PeMS-Bay (Li et al., 2018). These datasets capture diverse dynamics involving web activity, public health, social network interactions, and traffic forecasting. In all these tasks, the goal is to predict future node values from time-series data. We note that both the topology and node features evolve in Twitter Tennis, whereas in the other datasets only the latter change over time. For all experiments, we adhere to the data splits and experimental protocols established in the original works.

**Results.** Tables 4 and 5 report the forecasting performance of LAMP on the five tasks. Our model outperforms all baselines on ChickenPox Hungary and Wikipedia Math, and on par with the best-performing baselines on Twitter Tennis. On traffic forecasting benchmarks (i.e., PeMS-Bay and METR-LA), it achieves competitive performance, outperforming both temporal graph SSMs and classical temporal graph neural networks (such as DCRNN and Graph WaveNet), while remaining comparable to attention-based methods, which are known to incur higher computational cost. Overall, these results show that LAMP is highly competitive and suggest that explicitly modeling continuous-time dynamics with adaptive dissipativity is advantageous in real-world settings where filtering and preserving relevant information over spatio-temporal horizons is critical.

### 4.2.2. LINK-LEVEL PREDICTION

We evaluate LAMP on two popular link prediction datasets with evolving topology: AS-733 (Leskovec et al., 2005), representing the communication network of autonomous systems over a 26-month span derived from BGP logs; and

Bitcoin-$\alpha$ (Kumar et al., 2016; 2018), a "who-trusts-whom" network of Bitcoin traders. We use the experimental settings and data splits from Gravina & Bacciu (2024).

**Results.** We report the link prediction performance (F1 score) in Table 6 (additional metrics are reported in Appendix B.4). LAMP consistently achieves state-of-the-art results across both benchmarks, yielding an improvement of more than 8% on AS-733 compared to the strongest baseline. These results further reinforce the benefits of explicitly modeling the evolution of D-TDGs as a dynamical system that can adaptively balance conservative and non-conservative behaviors within its latent dynamics.

## 5. Conclusions

In this work, we introduce LAMP, a differential-equation-based GNN architecture for learning on discrete-time dynamic graphs (D-TDGs). LAMP models spatio-temporal propagation through a neural impulsive differential equation, combining conservative dynamics with a learned, channel-wise dissipative mechanism that adaptively regulates information retention over time and space.

We provide a theoretical analysis showing how this adaptive dissipativity enables stable latent evolution while preserving long-range propagation capabilities, allowing the model to selectively retain or attenuate information based on the task across both space and time, addressing key limitations of existing temporal GNNs in long-range settings.

Empirically, LAMP achieves strong performance on a newly proposed synthetic benchmark designed to stress-test long-range propagation jointly in space and time, as well as on multiple real-world node- and link-level prediction tasks.

*Table 5.* Forecasting errors (↓) on Metr-LA and PeMS-Bay (Horizon 12, Window 12). **First**, **second**, **third** best color-coded. Baselines from (Shao et al., 2022; Liu et al., 2023; Gao et al., 2024; Fan et al., 2024; Zhang et al., 2024; Ceni et al., 2025).

| Model | Metr-LA | | | PeMS-Bay | | |
|---|---|---|---|---|---|---|
| | MAE | RMSE | MAPE | MAE | RMSE | MAPE |
| HA | 6.99 | 13.89 | 17.54 | 3.31 | 7.54 | 7.65 |
| FC-LSTM | 4.37 | 8.69 | 14.00 | 2.37 | 4.96 | 5.70 |
| SVR | 6.72 | 13.76 | 16.70 | 3.28 | 7.08 | 8.00 |
| VAR | 6.52 | 10.11 | 15.80 | 2.93 | 5.44 | 6.50 |
| AdpSTGCN | 3.40 | 7.21 | 9.45 | 1.92 | 4.49 | 4.62 |
| ASTGCN | 6.51 | 12.52 | 11.64 | 2.61 | 5.42 | 6.00 |
| DCRNN | 3.60 | 7.60 | 10.50 | 2.07 | 4.74 | 4.90 |
| GMAN | 3.44 | 7.35 | 10.07 | 1.86 | 4.32 | 4.37 |
| Graph WaveNet | 3.53 | 7.37 | 10.01 | 1.95 | 4.52 | 4.63 |
| GTS | 3.46 | 7.31 | 9.98 | 1.95 | 4.43 | 4.58 |
| MTGNN | 3.49 | 7.23 | 9.87 | 1.94 | 4.49 | 4.53 |
| RGDAN | 3.26 | 7.02 | 9.73 | 1.82 | 4.20 | 4.28 |
| STAEformer | 3.34 | 7.02 | 9.70 | 1.88 | 4.20 | 4.41 |
| STD-MAE | 3.40 | 7.07 | 9.59 | 1.77 | 4.20 | 4.17 |
| PDFormer | 3.62 | 7.47 | 10.91 | 1.91 | 4.43 | 4.51 |
| STEP | 3.37 | 6.99 | 9.61 | 1.79 | 4.20 | 4.18 |
| STGCN | 4.59 | 9.40 | 12.70 | 2.49 | 5.69 | 5.79 |
| STSGCN | 5.06 | 11.66 | 12.91 | 2.26 | 5.21 | 5.40 |
| GGRNN | 3.88 | 8.14 | 10.59 | 2.34 | 5.14 | 5.21 |
| GraphSSM-S4 | 3.74 | 7.90 | 10.37 | 1.98 | 4.45 | 4.77 |
| LAMP | 3.48 | 7.43 | 10.65 | 1.89 | 4.39 | 4.48 |

*Table 6.* Test F1 (↑) scores on real-world link prediction tasks, averaged over 5 runs. **First**, **second**, and **third** best results are color-coded. Baselines results are from (Gravina & Bacciu, 2024).

| Model | AS-733 | Bitcoin $\alpha$ |
|---|---|---|
| DynGESN | $79.83_{\pm 5.27}$ | $69.98_{\pm 1.57}$ |
| EvolveGCN-H | $39.85_{\pm 34.24}$ | $29.55_{\pm 30.58}$ |
| EvolveGCN-O | $29.99_{\pm 37.10}$ | $31.74_{\pm 29.98}$ |
| GC-LSTM | $91.22_{\pm 0.13}$ | $91.22_{\pm 1.38}$ |
| LRGCN | $89.59_{\pm 0.33}$ | $91.33_{\pm 0.08}$ |
| LAMP | $96.20_{\pm 0.57}$ | $93.71_{\pm 0.09}$ |

Overall, LAMP consistently matches or outperforms state-of-the-art GNNs for D-TDGs, highlighting the importance of adaptive control over information retention in dynamic graph learning.

Future work will include extending the LAMP framework to irregularly sampled dynamic graphs (Gravina et al., 2024b), where observation times are not uniformly spaced; investigating alternative discretization strategies, such as adaptive multi-step integration schemes (Ascher & Petzold, 1998) ; and extending LAMP to higher-order graph shift operators (Defferrard et al., 2016; Wu et al., 2019a), which would decouple instantaneous spatial propagation from the ODE solver depth.

## Impact Statement

This paper presents work whose goal is to advance the field of Machine Learning. There are many potential societal consequences of our work, none of which we feel must be specifically highlighted here.

## Acknowledgements

FDC, AG, and DB acknowledge funding from EU-EIC EMERGE (Grant No. 101070918). ME acknowledges support from the Israeli Ministry of Innovation, Science & Technology.

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

# A. Proofs

## A.1. Jacobian Derivation and Spectral Properties

In this section, we formally derive the Jacobians for the continuous and discrete dynamics of LAMP and analyze the spectral properties of the neural vector field. We proceed in two steps: we first analyze the purely conservative variant of our LAMP obtained from Equation (2), and then the full version (i.e., Equation (7)), which incorporates the learnable diagonal damping $\mathbf{\Gamma}$. These results serve as the foundation for the sensitivity and stability proofs presented in the subsequent sections.

Let $\mathbf{h} = \text{vec}(\mathbf{H}(t)) \in \mathbb{R}^{nd}$ denote the vectorized latent state at time $t$, where we omit the time dependence for notational simplicity. Throughout the derivation, we use the identity $\text{vec}(\mathbf{A}\mathbf{X}\mathbf{B}) = (\mathbf{B}^{\top} \otimes \mathbf{A})\,\text{vec}(\mathbf{X})$, where $\otimes$ represents the Kronecker product.

### A.1.1. JACOBIAN OF THE CONSERVATIVE BACKBONE

We first consider the purely conservative vector field defined in Equation (2). Substituting the spatial operators $\mathcal{S}$ and $\mathcal{K}$ from Equations (3) and (4) into Equation (2), the dynamics can be written as:

$$\dot{\mathbf{H}} = \sigma\left(\mathbf{H}\mathbf{W}_{\text{sk}} - \mathbf{A}_{\text{sym}}\mathbf{H}\mathbf{V}_{\text{sk}} + \beta(\mathbf{A}_{\text{rw}} - \mathbf{A}_{\text{rw}}^{\top})\mathbf{H}\mathbf{Z}_{\text{sym}}\right). \tag{15}$$

Applying the vectorization operator yields:

$$\dot{\mathbf{h}} = \sigma\Big(\left(\mathbf{W}_{\text{sk}}^{\top} \otimes \mathbf{I}_n\right)\mathbf{h} - \left(\mathbf{V}_{\text{sk}}^{\top} \otimes \mathbf{A}_{\text{sym}}\right)\mathbf{h} + \beta\left(\mathbf{Z}_{\text{sym}}^{\top} \otimes (\mathbf{A}_{\text{rw}} - \mathbf{A}_{\text{rw}}^{\top})\right)\mathbf{h}\Big)$$
$$= \sigma\Big(\left((\mathbf{W}_{\text{sk}}^{\top} \otimes \mathbf{I}_n) - (\mathbf{V}_{\text{sk}}^{\top} \otimes \mathbf{A}_{\text{sym}}) + \beta(\mathbf{Z}_{\text{sym}}^{\top} \otimes (\mathbf{A}_{\text{rw}} - \mathbf{A}_{\text{rw}}^{\top}))\right)\mathbf{h}\Big). \tag{16}$$

We identify the total skew-symmetric weight matrix $\mathbf{M}_{\text{sk}}$ that aggregates the channel mixing and spatial diffusion components:

$$\mathbf{M}_{\text{sk}} = \underbrace{\mathbf{W}_{\text{sk}}^{\top} \otimes \mathbf{I}_n}_{\text{Channel Mixing}} - \underbrace{\mathbf{V}_{\text{sk}}^{\top} \otimes \mathbf{A}_{\text{sym}}}_{\text{Symmetric Diffusion}} + \beta\underbrace{\mathbf{Z}_{\text{sym}}^{\top} \otimes \left(\mathbf{A}_{\text{rw}} - \mathbf{A}_{\text{rw}}^{\top}\right)}_{\text{Antisymmetric Diffusion}}. \tag{17}$$

Since $\mathbf{M}_{\text{sk}}$ is a sum of skew-symmetric matrices (recalling that $\mathbf{W}_{\text{sk}}, \mathbf{V}_{\text{sk}}$ are skew-symmetric and $\mathbf{Z}_{\text{sym}}$ is symmetric), it satisfies $\mathbf{M}_{\text{sk}}^{\top} = -\mathbf{M}_{\text{sk}}$. The dynamics thus simplify to $\dot{\mathbf{h}} = \sigma(\mathbf{M}_{\text{sk}}\mathbf{h})$, and the Jacobian of the conservative vector field is obtained via the chain rule:

$$\mathbf{J}_{F_\theta}^{\text{cons}}(\mathbf{h}) = \frac{\partial \dot{\mathbf{h}}}{\partial \mathbf{h}} = \mathbf{D}(\mathbf{h})\mathbf{M}_{\text{sk}}, \tag{18}$$

where $\mathbf{D}(\mathbf{h}) = \text{diag}(\text{vec}(\sigma'(\mathbf{H})))$ contains the non-negative scalar derivatives of the activation function. As shown in (Gravina et al., 2025a), the product between $\mathbf{D}(\mathbf{h})$ and $\mathbf{M}_{\text{sk}}$ ensures

$$\text{Re}(\lambda(\mathbf{J}_{F_\theta}^{\text{cons}})) = 0. \tag{19}$$

This spectral property guarantees non-dissipativity, thus trajectories neither contract nor expand exponentially, so information injected at any time is preserved as it propagates through the latent state. This conservative behavior is the building block on top of which the adaptive damping is introduced.

### A.1.2. JACOBIAN WITH ADAPTIVE DAMPING

We now extend the derivation to the full vector field in Equation (7), which includes the learnable diagonal matrix $\mathbf{\Gamma}$. Repeating the substitution of $\mathcal{S}$ and $\mathcal{K}$ and the vectorization step, the dynamics become

$$\dot{\mathbf{h}} = \sigma\Big(\left((\mathbf{W}_{\text{sk}}^{\top} \otimes \mathbf{I}_n) - (\mathbf{V}_{\text{sk}}^{\top} \otimes \mathbf{A}_{\text{sym}}) + \beta(\mathbf{Z}_{\text{sym}}^{\top} \otimes (\mathbf{A}_{\text{rw}} - \mathbf{A}_{\text{rw}}^{\top})) - (\mathbf{\Gamma} \otimes \mathbf{I}_n)\right)\mathbf{h}\Big). \tag{20}$$

Reusing the skew-symmetric matrix $\mathbf{M}_{\text{sk}}$ from Equation (17), we define the total linear coupling matrix $\mathbf{M}$ as the sum of the conservative skew-symmetric term and the dissipative diagonal damping:

$$\mathbf{M} = \mathbf{M}_{\text{sk}} - \mathbf{\Gamma} \otimes \mathbf{I}_n. \tag{21}$$

The dynamics simplify to $\dot{\mathbf{h}} = \sigma(\mathbf{M}\mathbf{h})$. Therefore the Jacobian of the full continuous vector field is

$$\mathbf{J}_{F_\theta}(\mathbf{h}) = \frac{\partial \dot{\mathbf{h}}}{\partial \mathbf{h}} = \mathbf{D}(\mathbf{h})\mathbf{M} = \mathbf{D}(\mathbf{h})(\mathbf{M}_{\mathrm{sk}} - \mathbf{\Gamma} \otimes \mathbf{I}_n). \tag{22}$$

Here, $\mathbf{D}(\mathbf{h}) = \mathrm{diag}(\mathrm{vec}(\sigma'(\mathbf{H})))$ contains the non-negative scalar derivatives of the activation function, as in Section A.1.1.

A key property of this formulation is the explicit control over the spectrum of $\mathbf{J}_{F_\theta}$ induced by $\mathbf{\Gamma}$. Let $\lambda \in \mathbb{C}$ be an eigenvalue of $\mathbf{J}_{F_\theta}$ at any state $\mathbf{h}$. As shown in Section A.1.1, the skew-symmetric contribution $\mathbf{D}(\mathbf{h})\mathbf{M}_{\mathrm{sk}}$ alone places the spectrum on the imaginary axis. The diagonal term $-\mathbf{D}(\mathbf{h})(\mathbf{\Gamma} \otimes \mathbf{I}_n)$ shifts this spectrum into the left half-plane by a channel-wise amount proportional to the entries of $\mathbf{\Gamma}$ weighted by the activation derivatives:

$$-1 \le \mathrm{Re}(\lambda(\mathbf{J}_{F_\theta})) \le - \left( \min_k \sigma'_k \right) \cdot \lambda_{\min}(\mathbf{\Gamma}) \le 0. \tag{23}$$

Complementary to the continuous flow, the Jacobian of the discrete jump map $G_\phi$ in Equation (5) is linear:

$$\mathbf{J}_{G_\phi} = \frac{\partial \, \mathrm{vec}(\mathbf{H}(t_k^+))}{\partial \, \mathrm{vec}(\mathbf{H}(t_k^-))} = \frac{1}{\xi}\mathbf{I}_{nd}. \tag{24}$$

## A.2. Adaptive Memory Retention

**Theorem A.1** (Adaptive Memory Retention). *Let $\mathbf{G} = \mathbf{\Gamma} \otimes \mathbf{I}_n$. The instantaneous rate of change of LAMP, $\mu_2(\mathbf{J}_{F_\theta})$, is strictly bounded by:*

$$\mu_2(\mathbf{J}_{F_\theta}) \le \nu(\mathbf{h}) - \min_i \left( [\mathbf{D}(\mathbf{h})]_{ii}[\mathbf{G}]_{ii} \right), \tag{25}$$

*where the coupling expansion rate $\nu(\mathbf{h})$ is defined by the symmetric interaction between the activation derivatives and the mixing matrix:*

$$\nu(\mathbf{h}) = \lambda_{\max} \left( \frac{\mathbf{D}(\mathbf{h})\mathbf{M}_{sk} - \mathbf{M}_{sk}\mathbf{D}(\mathbf{h})}{2} \right). \tag{26}$$

*Consequently, strict contraction ($\mu_2 < 0$) is guaranteed whenever the dissipation is stronger than the expansion rate.*

*Proof.* We compute the symmetric part of the Jacobian $\mathbf{J}_{F_\theta} = \mathbf{D}(\mathbf{h})(\mathbf{M}_{\mathrm{sk}} - \mathbf{\Gamma} \otimes \mathbf{I}_n)$. For brevity, let $\mathbf{G} = \mathbf{\Gamma} \otimes \mathbf{I}_n$.

$$\begin{aligned} \frac{\mathbf{J}_{F_\theta} + \mathbf{J}_{F_\theta}^\top}{2} &= \frac{1}{2} \left( (\mathbf{D}(\mathbf{h})\mathbf{M}_{\mathrm{sk}} - \mathbf{D}(\mathbf{h})\mathbf{G}) + (\mathbf{D}(\mathbf{h})\mathbf{M}_{\mathrm{sk}} - \mathbf{D}(\mathbf{h})\mathbf{G})^\top \right) \\ &= \frac{1}{2} \left( \mathbf{D}(\mathbf{h})\mathbf{M}_{\mathrm{sk}} - \mathbf{D}(\mathbf{h})\mathbf{G} + \mathbf{M}_{\mathrm{sk}}^\top \mathbf{D}(\mathbf{h}) - \mathbf{G}^\top \mathbf{D}(\mathbf{h}) \right). \end{aligned} \tag{27}$$

Using the properties that $\mathbf{D}(\mathbf{h})$ and $\mathbf{G}$ are diagonal matrices, while $\mathbf{M}_{\mathrm{sk}}$ is skew-symmetric:

$$\begin{aligned} \frac{\mathbf{J}_{F_\theta} + \mathbf{J}_{F_\theta}^\top}{2} &= \frac{1}{2} \left( \mathbf{D}(\mathbf{h})\mathbf{M}_{\mathrm{sk}} - \mathbf{D}(\mathbf{h})\mathbf{G} - \mathbf{M}_{\mathrm{sk}}\mathbf{D}(\mathbf{h}) - \mathbf{G}\mathbf{D}(\mathbf{h}) \right) \\ &= \frac{1}{2} \left( \mathbf{D}(\mathbf{h})\mathbf{M}_{\mathrm{sk}} - \mathbf{M}_{\mathrm{sk}}\mathbf{D}(\mathbf{h}) - 2\mathbf{D}(\mathbf{h})\mathbf{G} \right) \\ &= \frac{1}{2} \left( \mathbf{D}(\mathbf{h})\mathbf{M}_{\mathrm{sk}} - \mathbf{M}_{\mathrm{sk}}\mathbf{D}(\mathbf{h}) \right) - \mathbf{D}(\mathbf{h})\mathbf{G}. \end{aligned} \tag{28}$$

Substituting this back into the logarithmic norm definition $\mu_2(\mathbf{J}_{F_\theta})$ and applying the sub-additivity property $\mu_2(\mathbf{A} + \mathbf{B}) \le \mu_2(\mathbf{A}) + \mu_2(\mathbf{B})$:

$$\begin{aligned} \mu_2(\mathbf{J}_{F_\theta}) &= \lambda_{\max} \left( \frac{\mathbf{D}(\mathbf{h})\mathbf{M}_{\mathrm{sk}} - \mathbf{M}_{\mathrm{sk}}\mathbf{D}(\mathbf{h})}{2} - \mathbf{D}(\mathbf{h})\mathbf{G} \right) \\ &\le \lambda_{\max} \left( \frac{\mathbf{D}(\mathbf{h})\mathbf{M}_{\mathrm{sk}} - \mathbf{M}_{\mathrm{sk}}\mathbf{D}(\mathbf{h})}{2} \right) + \lambda_{\max} \left( -\mathbf{D}(\mathbf{h})\mathbf{G} \right). \end{aligned} \tag{29}$$

Identifying the first term as the coupling expansion $\nu(\mathbf{h})$ and the second term as the negative of the minimum dissipation (using $\lambda_{\max}(-\mathbf{A}) = -\lambda_{\min}(\mathbf{A})$ for diagonal $\mathbf{A}$):

$$\mu_2(\mathbf{J}_{F_\theta}) \le \nu(\mathbf{h}) - \min_i \left( [\mathbf{D}(\mathbf{h})]_{ii}[\mathbf{G}]_{ii} \right). \tag{30}$$

$\square$

## A.3. Boundedness and Stability

**Proposition A.2** (Global Boundedness). *Assume bounded inputs such that $\sup_t \|\mathbf{X}(t)\|_F \leq B$ and a fixed sampling interval $\Delta t \approx \epsilon L$. Let $\omega = \sup_{\mathbf{h}} \mu_2(\mathbf{J}_{F_\theta})$ be the maximum logarithmic norm of the vector field over the sequence. If the memory normalization satisfies the stability condition $\xi > e^{\omega \epsilon L}$, the latent state $\mathbf{H}(t)$ remains globally bounded as $t \to \infty$, satisfying:*

$$\limsup_{t \to \infty} \|\mathbf{H}(t)\|_F \leq \frac{\mu \|\mathbf{U}\|_F B}{\xi - e^{\omega \epsilon L}}. \tag{31}$$

*Proof.* By the Coppel's Inequality (Coppel, 1965), for any continuous interval $[t_{k-1}, t_k)$, the state magnitude evolves according to the logarithmic norm of the Jacobian $\mathbf{J}_{F_\theta}$. Given $\omega = \sup_{\mathbf{h}} \mu_2(\mathbf{J}_{F_\theta})$, the growth is bounded by:

$$\|\mathbf{H}(t_k^-)\|_F \leq e^{\omega \epsilon L} \|\mathbf{H}(t_{k-1}^+)\|_F. \tag{32}$$

The stability of the system is determined by the interplay between this continuous evolution and the discrete normalization $\xi$ applied at each impulse. We define the recurrence for the post-jump state magnitude $y_k = \|\mathbf{H}(t_k^+)\|_F$. Substituting the continuous growth bound into the discrete jump equation:

$$\begin{aligned} y_k = \left\| \frac{1}{\xi} \left( \mathbf{H}(t_k^-) + \mu \mathbf{U} \mathbf{X}(t_k) \right) \right\|_F &\leq \frac{1}{\xi} \|\mathbf{H}(t_k^-)\|_F + \frac{\mu \|\mathbf{U}\|_F}{\xi} \|\mathbf{X}(t_k)\|_F \\ &\leq \frac{e^{\omega \epsilon L}}{\xi} y_{k-1} + \frac{\mu \|\mathbf{U}\|_F}{\xi} B. \end{aligned} \tag{33}$$

Let $\rho = \xi^{-1} e^{\omega \epsilon L}$ denote the contraction factor. The condition $\xi > e^{\omega \epsilon L}$ implies $\rho < 1$, establishing a contractive recurrence relation. Summing the resulting geometric series for $k \to \infty$, we obtain:

$$\limsup_{k \to \infty} y_k \leq \frac{\frac{\mu \|\mathbf{U}\|_F B}{\xi}}{1 - \rho} = \frac{\frac{\mu \|\mathbf{U}\|_F B}{\xi}}{1 - \frac{e^{\omega \epsilon L}}{\xi}}. \tag{34}$$

Multiplying the numerator and denominator by $\xi$ yields the final bound:

$$\limsup_{t \to \infty} \|\mathbf{H}(t)\|_F \leq \frac{\mu \|\mathbf{U}\|_F B}{\xi - e^{\omega \epsilon L}}. \tag{35}$$

$\square$

## A.4. Sensitivity Bound

**Theorem A.3** (Sensitivity Bound). *Let $[s, t] \subseteq [t_1, t_T]$ be an arbitrary sub-interval of the input sequence times and let $s \leq t_i, \ldots, t_j \leq t \in \mathcal{T}$ be the discrete times at which impulses occur in the interval. Then, the LAMP sensitivity of node $v$ with respect to node $u$ is bounded by:*

$$\left\| \frac{\partial \mathbf{h}_v(t)}{\partial \mathbf{h}_u(s)} \right\|_2 \leq \underbrace{\xi^{-(j-i+1)} (c_\sigma w d)^{t-s}}_{model} \underbrace{\mathbf{\Psi}_{uv}}_{topology}. \tag{36}$$

*Here, $c_\sigma$ is the Lipschitz constant of the activation $\sigma$ and $w \in \mathbb{R}$ is the maximal entry-value over all weight matrices. The matrix $\mathbf{\Psi} \in \mathbb{R}^{n \times n}$ is the chain multiplication of each message passing matrix adopted at impulse times:*

$$\mathbf{\Psi} = \mathbf{S}_{j+1}^{t-t_j} \left[ \prod_{k=j}^{i+1} \mathbf{S}_k^{\Delta t_k} \right] \mathbf{S}_i^{t_i - s},$$

$$\mathbf{S}_k = \mathbf{I} + \mathbf{A}_{sym}(t_k) + \beta \left( \mathbf{A}_{rw}(t_k) - (\mathbf{A}_{rw}(t_k))^\top \right).$$

*Proof.* By applying the chain rule over the sequence of impulse times $t_i, \ldots, t_j$ in the interval $[s, t]$, we express this derivative as a product of Jacobians across continuous flows and discrete updates. To capture the total node-to-node influence, we sum over all possible sequences of intermediate nodes $p_k$ through the network:

$$\frac{\partial \mathbf{h}_v(t)}{\partial \mathbf{h}_u(s)} = \sum_{p_j, \ldots, p_i} \frac{\partial \mathbf{h}_v(t)}{\partial \mathbf{h}_{p_j}(t_j^+)} \left[ \prod_{k=j}^{i+1} \frac{\partial \mathbf{h}_{p_k}(t_k^+)}{\partial \mathbf{h}_{p_k}(t_k^-)} \frac{\partial \mathbf{h}_{p_k}(t_k^-)}{\partial \mathbf{h}_{p_{k-1}}(t_{k-1}^+)} \right] \frac{\partial \mathbf{h}_{p_i}(t_i^-)}{\partial \mathbf{h}_u(s)}$$

$$= \sum_{p_j, \ldots, p_i} \frac{\partial \mathbf{h}_v(t)}{\partial \mathbf{h}_{p_j}(t_j^+)} \left[ \prod_{k=j}^{i+1} [\mathbf{J}_{G_\phi}]_{p_k p_k} \frac{\partial \mathbf{h}_{p_k}(t_k^-)}{\partial \mathbf{h}_{p_{k-1}}(t_{k-1}^+)} \right] \frac{\partial \mathbf{h}_{p_i}(t_i^-)}{\partial \mathbf{h}_u(s)} \tag{37}$$

We analyze the discrete and continuous components of this sum separately. First, at each impulse time $t_k$, we use the Jacobian of the impulse map of Equation (24), which is diagonal. This implies that the jumps do not mix information between nodes, but simply scale the gradients uniformly by $\xi^{-1}$, independently of any node.

Second, for any continuous interval $(t_a, t_b)$ between impulses, we re-use the structural bound from (Gravina et al., 2025a), noting that we have constant residual and aggregation terms $c_a = c_b = 1$. The sensitivity between intermediate nodes $p$ and $q$ is bounded by the corresponding entry in the structural operator $\mathbf{S}$:

$$\left\| \frac{\partial \mathbf{h}_p(t_b)}{\partial \mathbf{h}_q(t_a)} \right\|_2 \leq (c_\sigma w d)^{t_b - t_a} [\mathbf{S}^{t_b - t_a}]_{pq}, \tag{38}$$

$$= (c_\sigma w d)^{t_b - t_a} \left[ \left( \mathbf{I} + \mathbf{A}_{\text{sym}} + \beta \left( \mathbf{A}_{\text{rw}} - (\mathbf{A}_{\text{rw}})^\top \right) \right)^{t_b - t_a} \right]_{pq}.$$

Substituting the impulse Jacobian from Equation (24) and the structural flow bounds from Equation (38) back into the chain rule expansion, we proceed to bound the total sum. We apply the triangle inequality to the sum over all intermediate nodes $p_k \in \mathcal{V}$. We perform the algebraic simplification in steps, first grouping the scalar constants and then factoring them out of the node summation:

$$\left\| \frac{\partial \mathbf{h}_v(t)}{\partial \mathbf{h}_u(s)} \right\|_2 \leq \sum_{p_j \in \mathcal{V}} \cdots \sum_{p_i \in \mathcal{V}} \left( \frac{1}{\xi} (c_\sigma w d)^{t - t_j} [\mathbf{S}_{j+1}^{t - t_j}]_{v p_j} \right) \prod_{k=j}^{i+1} \left( \frac{1}{\xi} (c_\sigma w d)^{\Delta t} [\mathbf{S}_k^{\Delta t}]_{p_k p_{k-1}} \right) (c_\sigma w d)^{t_i - s} [\mathbf{S}_i^{t_i - s}]_{p_i u}$$

$$= \sum_{p_j \in \mathcal{V}} \cdots \sum_{p_i \in \mathcal{V}} \left( \frac{1}{\xi} \prod_{k=j}^{i+1} \frac{1}{\xi} \right) \left( (c_\sigma w d)^{t - t_j} \prod_{k=j}^{i+1} (c_\sigma w d)^{\Delta t} (c_\sigma w d)^{t_i - s} \right) \left( [\mathbf{S}_{j+1}^{t - t_j}]_{v p_j} \cdots [\mathbf{S}_i^{t_i - s}]_{p_i u} \right)$$

$$= \left( \frac{1}{\xi} \prod_{k=j}^{i+1} \frac{1}{\xi} \right) \left( (c_\sigma w d)^{t - t_j + \sum \Delta t + t_i - s} \right) \left( \sum_{p_j \in \mathcal{V}} \cdots \sum_{p_i \in \mathcal{V}} [\mathbf{S}_{j+1}^{t - t_j}]_{v p_j} \cdots [\mathbf{S}_k^{\Delta t}]_{p_k p_{k-1}} \cdots [\mathbf{S}_i^{t_i - s}]_{p_i u} \right)$$

$$= \left( \xi^{-1} \cdot \xi^{-(j-i)} \right) (c_\sigma w d)^{t - s} \left( \sum_{p_j \in \mathcal{V}} \cdots \sum_{p_i \in \mathcal{V}} [\mathbf{S}_{j+1}^{t - t_j}]_{v p_j} \cdots [\mathbf{S}_k^{\Delta t}]_{p_k p_{k-1}} \cdots [\mathbf{S}_i^{t_i - s}]_{p_i u} \right)$$

$$= \xi^{-(j-i+1)} (c_\sigma w d)^{t - s} \left[ \mathbf{S}_{j+1}^{t - t_j} \cdot \left( \prod_{k=j}^{i+1} \mathbf{S}_k^{\Delta t} \right) \cdot \mathbf{S}_i^{t_i - s} \right]_{vu}$$

$$= \xi^{-(j-i+1)} (c_\sigma w d)^{t - s} [\boldsymbol{\Psi}]_{vu}. \tag{39}$$

This confirms the bound stated in the theorem. $\qquad \square$

## B. Additional Results and Comparisons

In this section, we provide empirical verification of the theoretical properties derived in Section 2.3, specifically the Adaptive Retention (Theorem 2.1) and Input-to-State Stability (Proposition 2.2). We first demonstrate that the eigenvalues of the Jacobian fall in the stable left half of the plane, and then we conduct three distinct analyses to demonstrate how LAMP manages the trade-off between stability and memory capacity in practice.

## B.1. Spectral Analysis of LAMP

We provide an extended analysis of the properties induced by the definition of LAMP by studying the spectrum of the model's Jacobian. Specifically, we take the best weights of LAMP after training on ChickenPox, and plot the Jacobian eigenvalues at the first three snapshots of the sequence in Figure 2.

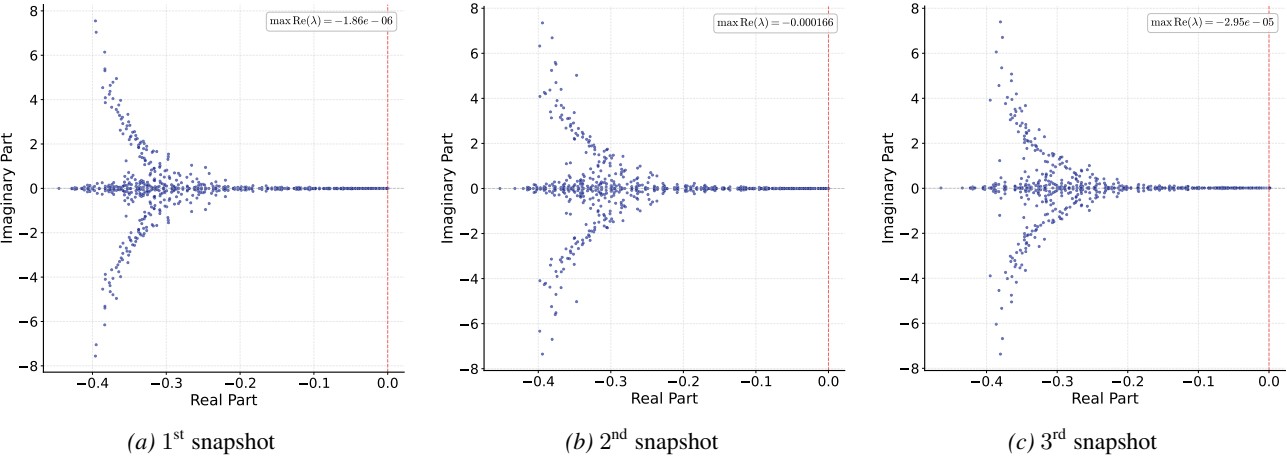

*(a)* $1^{\text{st}}$ snapshot          *(b)* $2^{\text{nd}}$ snapshot          *(c)* $3^{\text{rd}}$ snapshot

*Figure 2.* Evolution of LAMP's Jacobian eigenvalues across the first three snapshots on the ChickenPox dataset.

The figure displays all Jacobian's eigenvalues. This visually confirms that they strictly reside in the stable left-half plane. Furthermore, the figure shows that the real parts of the eigenvalues, which govern stability and adaptive memory retention, are bounded below by $-0.5$. This lower bound directly correlates with the theory in Equation (23) and the learned values of the diagonal damping matrix $\mathbf{\Gamma}$ (0.6194 on average).

## B.2. Global Stability and Boundedness

We quantify the adaptability of the dissipative mechanism to the task via the *stability ratio* $R_t$, defined as the competition between the instantaneous rate of expansion or contraction (i.e., the logarithmic norm of the Jacobian $\mu_2(\mathbf{J}_{F_\theta})$) and the discrete memory normalization $\xi$:

$$R_t = \frac{e^{\mu_2(\mathbf{J}_{F_\theta}(\mathbf{h}(t)))\epsilon L}}{\xi}. \tag{40}$$

The condition $R_t < 1$ implies strict linear contraction, a sufficient condition for stability that typically entails exponential memory decay. Conversely, $R_t \gtrsim 1$ indicates a regime where linear stability is violated locally to preserve information. We validate these distinct operating modes by comparing the inferred dynamics on a real-world task, ChickenPox, against a chosen task requiring long-range dependency modeling by design, namely the three-colors setting of `ColoredLeafCounting` with $T = 400$. We show that instability arises when the ratio increases over time.

As illustrated in Figure 3, the model exhibits a sharp phase transition depending on the task. On the ChickenPox dataset (Figure 3a), the system maintains a contractive profile with $R_t \approx 0.65$, since the contraction is stronger than the expansion. This confirms that, when needed, the model defaults to a controlled, contractive dynamic capable of filtering out irrelevant data. In contrast, for the long-range task (Figure 3b), the dynamics shift into a conservation regime. Here, conservation refers to a state where the model minimizes internal dissipation to sustain the hidden state magnitude near its contractive boundary ($R_t \lesssim 1$), effectively building up relevant information. The system relies on its nonlinearities to dampen internal expansion, preserving the hidden state's information across the sequence without requiring the dynamics to be strictly contractive. This balance ensures that memory remains persistent yet stable, showcasing $\mathbf{\Gamma}$'s ability to adapt the system's dissipation to the temporal demands of the task.

To ensure that operating in this conservation regime does not compromise global stability, we verify the Input-to-State Stability (ISS) property discussed in Proposition 2.2. ISS requires that the system's state magnitude $\|\mathbf{H}(\infty)\|_F$ remains bounded by a function of the constant input magnitudes $\|\mathbf{X}(t_k)\|_F$. We simulate the system response when perturbing each input at impulse time $t_k$ by a factor $\alpha \in [1, 50]$. Figure 4 shows the linear relationship between the controlled perturbation of

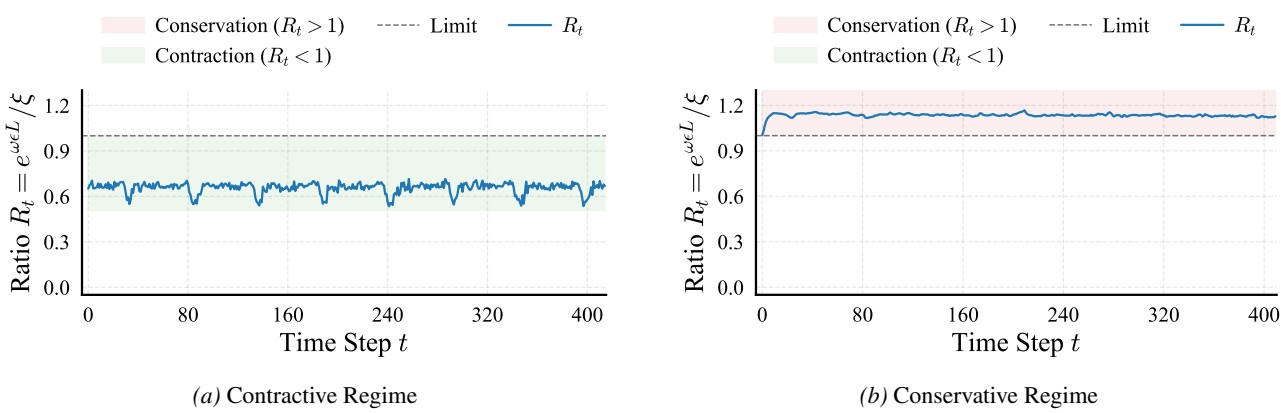

*(a)* Contractive Regime          *(b)* Conservative Regime

*Figure 3.* (a) On the ChickenPox dataset, the model maintains a safe stability ratio ($R_t \approx 0.65$), actively dissipating history to prioritize recent signal. (b) On the the three-colors setting of `ColoredLeafCounting` (i.e., a long-range task), the model reaches a state preserving information in its memory ($R_t \gtrsim 1$) without exploding.

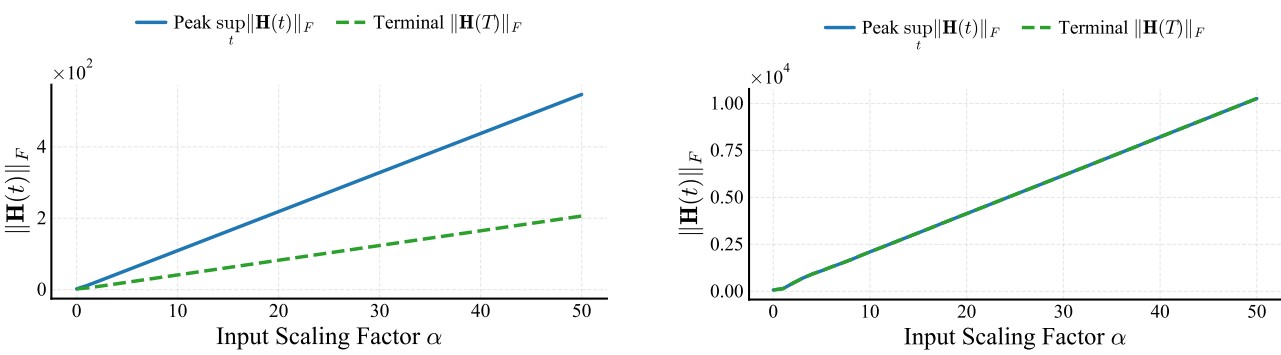

*(a)* Input-to-State Stability of LAMP on the ChickenPox dataset.     *(b)* Input-to-State Stability of LAMP on the three-colors `ColoredLeafCounting` task with $T = 400$.

*Figure 4.* ISS of LAMP on two selected datasets, i.e., ChickenPox and the three-colors `ColoredLeafCounting`.

the input signal and the magnitude of the hidden state at the final timestep of each sequence, as expected from Equation (13). In Figure 4a, we observe that system contraction causes the latent state to reach its maximum magnitude at an intermediate timestep, rather than at the final step. In contrast, Figure 4b suggests that LAMP integrates information until the sequence concludes, given that $\|\mathbf{H}(400)\|_F$ exhibits peak magnitude at that point.

This empirical evidence shows that our model learns to maintain its latent state magnitude bounded. In particular, LAMP exhibits a bounded growth rate (measured by $\mu(\mathbf{J}_{F_\theta})$) in tasks having a different nature. Moreover, this analysis showed that our model satisfies the ISS property, even under large perturbations to the input features.

Lastly, we observe that in all the experiments in Section 4 we restricted our search to $\xi \in \{1, 2\}$ to to ensure system stability, avoiding $\xi < 1$ to prevent energy amplification.

## B.3. Runtimes

We analyze the training and inference runtimes of LAMP compared to other methods by collecting the mean and standard deviation of execution times over 10 selected epochs on the `ColoredLeafCounting` task, restricted to the three-color setting (refer to Table 10 for graph size details). For computational fairness, all models are evaluated on a single NVIDIA V100 GPU with a fixed batch size of 64 to assess how runtime scales with increasing sequence length $T$. The first epoch is excluded to ensure that GPU memory allocation overhead is not counted in the statistics. As shown in Table 7, we observe distinct trade-offs between computational efficiency and predictive performance across the evaluated methods.

Regarding computational efficiency, **DynGESN**, **EvolveGCN-O**, and **EvolveGCN-H** consistently demonstrate the lowest

training and inference latencies across all sequence lengths ($T$). These methods effectively minimize the computational overhead, likely due to their lighter architectural designs. Our proposed LAMP immediately follows the fastest models, aligning closely with **A3T-GCN** and **T-GCN**; it scales linearly with sequence length but remains significantly faster than the heavier recurrent baselines like **DCRNN**, **GC-LSTM**, and **GCRN-GRU**.

However, the advantages of the fastest models diminish when analyzing predictive accuracy. While **DynGESN** provides the fastest execution, it fails to capture the complex temporal dynamics required for this task, resulting in high MSE scores comparable to the lower-performing baselines. In contrast, LAMP achieves a massive reduction in error, outperforming the second-best method (**DCRNN**) by a substantial margin in terms of Test MSE (for instance, 0.057 vs. 1.283 at $T = 200$). Furthermore, LAMP achieves this superior performance while requiring approximately one-third of the training time of DCRNN (10.86s vs. 29.71s at $T = 200$). This indicates that LAMP offers the most favorable balance, delivering state-of-the-art accuracy without the prohibitive computational costs associated with other state-of-the-art models.

*Table 7.* Mean and standard deviation of the training and inference runtime (seconds) and obtained MSE on the `ColoredLeafCounting` task for the three-color setting. All metrics are shown by increasing sequence length $T$. **First**, **second**, and **third** best results are color-coded, for each metric.

| Metrics | Method | Sequence Length | | |
|---|---|---|---|---|
| | | **T = 200** | **T = 400** | **T = 600** |
| Epoch Time (s) | | $11.35 \pm 0.33$ | $25.17 \pm 1.49$ | $35.36 \pm 2.80$ |
| Inf. Time (s) | A3T-GCN | $1.14 \pm 0.01$ | $2.53 \pm 0.23$ | $3.52 \pm 0.27$ |
| Test MSE | | $1.304 \pm 0.009$ | $1.003 \pm 0.002$ | $0.836 \pm 0.003$ |
| Epoch Time (s) | | $29.71 \pm 0.92$ | $58.46 \pm 3.57$ | $84.75 \pm 0.36$ |
| Inf. Time (s) | DCRNN | $2.24 \pm 0.02$ | $4.60 \pm 0.38$ | $6.58 \pm 0.01$ |
| Test MSE | | **1.283** $\pm0.023$ | **0.811** $\pm0.177$ | **0.675** $\pm0.043$ |
| Epoch Time (s) | | **6.03** $\pm0.29$ | **12.46** $\pm0.70$ | **18.36** $\pm0.92$ |
| Inf. Time (s) | DynGESN | **0.74** $\pm0.05$ | **1.52** $\pm0.11$ | **2.24** $\pm0.13$ |
| Test MSE | | $1.303 \pm 0.008$ | $1.010 \pm 0.008$ | $0.838 \pm 0.002$ |
| Epoch Time (s) | | **8.44** $\pm0.15$ | **16.58** $\pm0.32$ | **25.74** $\pm0.36$ |
| Inf. Time (s) | EvolveGCN-H | **1.03** $\pm0.00$ | **2.00** $\pm0.03$ | **3.13** $\pm0.03$ |
| Test MSE | | $1.305 \pm 0.017$ | $0.991 \pm 0.000$ | $0.868 \pm 0.001$ |
| Epoch Time (s) | | **7.49** $\pm0.36$ | **13.91** $\pm0.84$ | **25.69** $\pm1.12$ |
| Inf. Time (s) | EvolveGCN-O | **0.87** $\pm0.06$ | **1.63** $\pm0.11$ | **2.96** $\pm0.18$ |
| Test MSE | | $1.307 \pm 0.018$ | **0.991** $\pm0.000$ | $0.868 \pm 0.001$ |
| Epoch Time (s) | | $19.36 \pm 0.37$ | $32.99 \pm 0.45$ | $53.87 \pm 3.84$ |
| Inf. Time (s) | GC-LSTM | $1.90 \pm 0.01$ | $3.18 \pm 0.01$ | $5.32 \pm 0.52$ |
| Test MSE | | $1.293 \pm 0.020$ | $1.004 \pm 0.013$ | $0.837 \pm 0.004$ |
| Epoch Time (s) | | $12.89 \pm 0.24$ | $38.15 \pm 0.79$ | $57.33 \pm 1.67$ |
| Inf. Time (s) | GCRN-GRU | $1.45 \pm 0.01$ | $3.83 \pm 0.07$ | $5.70 \pm 0.05$ |
| Test MSE | | $1.303 \pm 0.003$ | $1.003 \pm 0.010$ | **0.830** $\pm0.017$ |
| Epoch Time (s) | | $25.54 \pm 0.80$ | $47.43 \pm 0.77$ | $46.92 \pm 0.61$ |
| Inf. Time (s) | GCRN-LSTM | $2.52 \pm 0.19$ | $4.61 \pm 0.03$ | $5.11 \pm 0.08$ |
| Test MSE | | **1.248** $\pm0.078$ | $1.012 \pm 0.008$ | $0.837 \pm 0.002$ |
| Epoch Time (s) | | $11.06 \pm 0.42$ | $22.93 \pm 1.82$ | $32.25 \pm 0.53$ |
| Inf. Time (s) | T-GCN | $1.15 \pm 0.07$ | $2.34 \pm 0.23$ | $3.38 \pm 0.23$ |
| Test MSE | | $1.307 \pm 0.010$ | $1.004 \pm 0.005$ | $0.838 \pm 0.002$ |
| Epoch Time (s) | | $10.86 \pm 0.16$ | $21.59 \pm 0.25$ | $31.80 \pm 0.12$ |
| Inf. Time (s) | LAMP (our) | $1.18 \pm 0.02$ | $2.33 \pm 0.00$ | $3.49 \pm 0.01$ |
| Test MSE | | **0.057** $\pm0.015$ | **0.264** $\pm0.199$ | **0.223** $\pm0.071$ |

## B.4. Extended Evaluation Metrics

In this section we report additional evaluation metrics (i.e., AUROC and F1 score) for link prediction in Table 8. Even under different metrics, LAMP consistently maintains state-of-the-art performance.

*Table 8.* Results for link prediction on real world datasets. **First**, **second**, and **third** best results are color-coded.

| Model | AS-733 | | Bitcoin $\alpha$ | |
|---|---|---|---|---|
| | AUROC ↑ | F1 ↑ | AUROC ↑ | F1 ↑ |
| DynGESN | $95.34_{\pm 0.04}$ | $79.83_{\pm 5.27}$ | $97.68_{\pm 0.12}$ | $69.98_{\pm 1.57}$ |
| EvolveGCN-H | $59.52_{\pm 17.53}$ | $39.85_{\pm 34.24}$ | $51.35_{\pm 2.88}$ | $29.55_{\pm 30.58}$ |
| EvolveGCN-O | $58.90_{\pm 17.80}$ | $29.99_{\pm 37.10}$ | $51.42_{\pm 2.84}$ | $31.74_{\pm 29.98}$ |
| GC-LSTM | $96.35_{\pm 0.01}$ | $91.22_{\pm 0.13}$ | $97.75_{\pm 0.17}$ | $91.22_{\pm 1.38}$ |
| LRGCN | $94.77_{\pm 0.23}$ | $89.59_{\pm 0.33}$ | $98.05_{\pm 0.03}$ | $91.33_{\pm 0.08}$ |
| LAMP | $99.00_{\pm 0.22}$ | $96.20_{\pm 0.57}$ | $98.24_{\pm 0.03}$ | $93.71_{\pm 0.09}$ |

## B.5. Ablations

We study the sensitivity of LAMP to its hyperparameters $\epsilon$, $L$ and $\beta$. Starting from the best configuration on Twitter Tennis, we perform a grid search over each hyperparameter while fixing the others to their optimal values. Test MSE is reported in Table 9.

*Table 9.* Performance of LAMP on Twitter Tennis, when varying one among $\epsilon$, $L$, and $\beta$, and fixing the others to their optimal values.

| $\epsilon$ | 0.01 | 0.1 | 0.5 | 1.0 | |
|---|---|---|---|---|---|
| MSE | 0.3003 | 0.2985 | 0.3415 | 0.3359 | |

| $L$ | 1 | 2 | 4 | 8 | 16 |
|---|---|---|---|---|---|
| MSE | 0.3065 | 0.3033 | 0.3035 | 0.2985 | 0.2995 |

| $\beta$ | -1.0 | -0.1 | 0.0 | 0.1 | 1.0 |
|---|---|---|---|---|---|
| MSE | 0.2985 | 0.3036 | 0.3072 | 0.3026 | 0.3005 |

The step size $\epsilon$ is the most influential hyperparameter: performance is stable for small values ($\epsilon \leq 0.1$) but degrades markedly for larger ones ($\epsilon \geq 0.5$), with MSE rising from 0.2985 to 0.3415. This mirrors the behavior of classical numerical solvers for differential equations, where large step sizes introduce integration error and instability. In contrast, performance is largely insensitive to both the number of layers/iterations $L$ and the parameter $\beta$: across their respective ranges the MSE varies by less than 0.01, indicating the model is robust to these choices.

## C. Experimental Details

In this section, we provide additional details regarding our experiments, including employed baselines, datasets, and hyperparameter space.

### C.1. Employed Baselines

To evaluate the effectiveness of LAMP, we compare it against a comprehensive set of baselines. Specifically:

**A3T-GCN** (Bai et al., 2021), **AdpSTGCN** (Zhang et al., 2024), **ASTGCN** (Guo et al., 2019), **DCRNN** (Li et al., 2018), **DynGESN** (Micheli & Tortorella, 2022), **DyGrAE** (Taheri & Berger-Wolf, 2020), **EvolveGCN-H** and **EvolveGCN-O** (Pareja et al., 2020), **FC-LSTM** (Sutskever et al., 2014), **GC-LSTM** (Chen et al., 2022), **GCRN-GRU** and **GCRN-LSTM** (Seo et al., 2018), **GMAN** (Zheng et al., 2020), **Graph WaveNet** (Wu et al., 2019b), **GTS** (Shang et al., 2021), **HMM4G** (Errica et al., 2023), **LRGCN** (Li et al., 2019), **MPNN-LSTM** (Gilmer et al., 2017), **MTGNN** (Wu et al., 2020), **RGDAN** (Fan et al., 2024), **STAEformer** (Liu et al., 2023), **PDformer** (Jiang et al., 2023), **STD-MAE** (Gao et al., 2024), **STEP**

(Shao et al., 2022), **STGCN** (Yu et al., 2018), **STSGCN** (Sofianos et al., 2021), **T-GCN** (Zhao et al., 2020), **GGRNN** (Ruiz et al., 2020), **GraphSSM** (Li et al., 2024), **HA** (Historical Average), **SVR** (Smola & Schölkopf, 2004), and **VAR** (Lu et al., 2016).

### C.2. Employed Datasets

In our experiments, we evaluate LAMP on a variety of dynamic graph benchmarks and a novel spatio-temporal task. We utilize the `ColoredLeafCounting` benchmark to test joint spatio-temporal propagation over long horizons ($T = 200$ to 600). For node-level forecasting, we employ **Wikipedia Math**, which tracks daily page views; **ChickenPox Hungary**, representing epidemiological spread across counties; **Twitter Tennis**, modeling social interaction intensity; and the large-scale traffic datasets **Metr-LA** and **PeMS-Bay**, which require capturing fine-grained correlations in sensor networks. Finally, for link prediction, we use **AS-733**, a 26-month span of BGP logs representing autonomous system communications, and **Bitcoin** $\alpha$, which captures evolving trust relationships between cryptocurrency traders. In Table 10, we report the statistics of the employed datasets.

*Table 10.* Employed datasets statistics

| Dataset | #Graphs | #Nodes | #Edges | Sequence Length | Evolving Topology |
|---|---|---|---|---|---|
| **Twitter Tennis** | 1 | 1000 | 40839 | 120 | ✓ |
| **ChickenPox** | 1 | 20 | 102 | 517 | ✗ |
| **Wikipedia Math** | 1 | 1068 | 27079 | 723 | ✗ |
| **Bitcoin** $\alpha$ | 1 | 3783 | 24186 | 24186 | ✓ |
| **AS-733** | 1 | 7716 | 13895 | 733 | ✓ |
| **Metr-LA** | 1 | 207 | 1515 | 34272 | ✗ |
| **PeMS-Bay** | 1 | 325 | 2369 | 52116 | ✗ |
| `ColoredLeafCounting` | 600 | 51 | 50 | 200, 400, 600 | ✗ |

### C.3. Hyperparameter Space

The hyperparameter space employed by LAMP across all tasks is reported in Table 11.

*Table 11.* The grid of hyperparameters employed by LAMP during model selection for all evaluated tasks. Datasets include: Traffic Forecasting (**PeMS-Bay**, **Metr-LA**), Node Prediction (**ChickenPox**, **Wikipedia Math**, **Twitter Tennis**), and Link Prediction (**AS-733**, **Bitcoin** $\alpha$).

| Hyperparameters | Values | | | |
|---|---|---|---|---|
| | **Traffic Forecasting** | **Node Prediction** | **Link Prediction** | `ColoredLeafCounting` |
| Optimizer | Adam | Adam | Adam | Adam |
| Learning rate | 0.001, 0.0001 | 0.01, 0.001 | 0.01, 0.001 | 0.01, 0.001 |
| Weight decay | 0 | $10^{-4}$ | $10^{-4}$ | $10^{-4}$ |
| $d$ | 512 | 32, 64, 128 | 32, 64, 128 | 32 |
| $L$ | window + horizon | 1, 3 | 1, 3 | 1 |
| $\gamma$ | 0.1 | 0.01, 0.1 | 0.01, 0.1 | 0.01, 0.1 |
| $\epsilon$ | 0.01, 0.05, 0.1, 0.3 | 0.1, 0.15, 0.2, 0.25, 0.3 | 0.1 | 0.1 |
| $\beta$ | 0.1 | -1, -0.5, 0.1, 0.5, 1 | -0.5, 0.1, 0.5 | -1, 0.1, 1 |
| $\mu$ | 1 | 1 | 1 | 1 |
| $\xi$ | 1 | 1, 2 | 1, 2 | 1 |
| $\sigma$ | tanh | tanh | tanh | tanh |

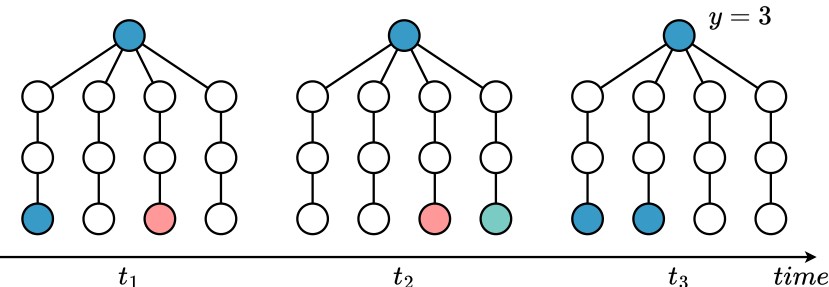

*Figure 5.* Illustration of the colored leaf-counting task over 3 timesteps on a rooted-tree of depth 3 and 4 branches. The root and all leaf nodes are assigned with a color from the set red, green, blue. At each timestep, leaf nodes may emit a binary signal ("blink") that is propagated through the graph. At the end of the sequence, the root node must count the total number of blinks produced by leaves that share its color. In the figure, the number of blinks from leaf nodes sharing the same root color is $y = 3$.

### C.4. ColoredLeafCounting Benchmark

In this section, we introduce the `ColoredLeafCounting` benchmark, a novel synthetic task specifically designed to stress-test effective long-range information propagation across both space and time.

The benchmark is defined over sequences of graph snapshots whose topology is a rooted tree, with all (i.e., five) leaf nodes located at a fixed distance of 10 hops from the root. For illustrative purposes, Figure 5 depicts an example consisting of a rooted tree with four leaf nodes at distance three from the root. The root node and all leaf nodes are randomly assigned a color from a predefined set.

Over a sequence of discrete timesteps, leaf nodes may emit a binary signal ("blink") that is propagated through the graph. Each leaf node independently emits a blink at each timestep with probability $0.5$. When a leaf node emits a blink, its input feature is given by the one-hot encoding of its assigned color, while all other nodes receive a zero vector. The root node, instead, receives a constant input encoding its target color throughout the sequence.

At the final timestep $T$, the learning objective is to predict the total number of blinks emitted over time by leaf nodes whose color matches that of the root. Successfully solving this task requires the model to propagate information over long spatial paths (from leaves to root) while simultaneously aggregating signals over extended temporal horizons, selectively filtering irrelevant inputs, and accumulating only those that are both temporally and semantically relevant. Figure 5 illustrate the `ColoredLeafCounting` benchmark.

We consider three different sequence lengths, $T \in \{200, 400, 600\}$, and two color configurations: a binary setting black, white and a three-class setting red, green, blue. These settings induce increasing levels of difficulty by extending the temporal aggregation horizon and increasing the diversity of node attributes to be filtered out. Overall, this results in a total of six distinct task configurations. For each configuration, we consider a dataset of 600 sequences.

All models are trained using an MSE loss, Adam optimizer with a maximum of 250 epochs and early stopping patience of 50 epochs, following an 80/10/10 train/validation/test split. Reported results are averaged over five random initializations.

