# OpenReview forum: "Adaptive Memory Retention in Dynamic Graphs"
_ICML.cc/2026/Conference — ICML 2026 spotlight_

### Official Review · Reviewer_uuzf · 2026-02-25

**Soundness:** 4
**Presentation:** 2
**Significance:** 3
**Originality:** 3
**Overall Recommendation:** 5
**Confidence:** 5

**Summary:**

This paper tackles long-term memory loss in Discrete-Time Dynamic Graphs (DTDGs) caused by eigenvalue accumulation and over-smoothing by modeling spatio-temporal dynamics as a continuous ODE system. The core innovation lies in decomposing the graph operator into symmetric and skew-symmetric components: it preserves the skew-symmetric part for conservative, non-dissipative information flow while stabilizing the symmetric part using an input-agnostic adaptive damping term (forget gate). This targeted control prevents signal explosion without sacrificing memory capacity, theoretically guaranteeing stability and achieving state-of-the-art performance on multiple long-range spatio-temporal benchmarks.

**Compliance With Llm Reviewing Policy:**

Affirmed.

**Final Justification:**

This is a relatively solid theoretical work. My previous concerns mainly focused on the effective module and the writing. Given that the authors' response has resolved these issues, I have decided to raise my score to 5.

**Key Questions For Authors:**

**Q1** : Given that modern State Space Models (e.g., Mamba) and Gated Linear Attention mechanisms have demonstrated the superiority of input-dependent gating for content-aware long-range modeling, why does this work employ a static, input-agnostic diagonal damping matrix ($\Gamma$), and does this design choice prioritize theoretical tractability over the ability to adapt to instantaneous noise in non-stationary dynamic graphs?

**Q2** : While the paper provides rigorous mathematical proofs (Theorems 2.1--2.3) that the symmetric decomposition and damping term constrain the system Jacobian's eigenvalues to the stable left-half plane, can the authors provide spectral analysis case studies (e.g., eigenvalue spectrum plots or trajectory phase portraits) to visually demonstrate this ``eigenvalue suppression'' and bridge the gap between abstract control-theory bounds and empirical stability?

**Q3** : Since the current analysis focuses on first-order adjacency operators, can the proposed ``decompose-and-control'' framework and the adaptive damping mechanism $\Gamma$ be theoretically and empirically extended to higher-order graph kernels (e.g., $A^2$ or PageRank matrices), where eigenvalue squaring ($\lambda \to \lambda^2$) might amplify instability beyond the capacity of a simple diagonal dampener?

**Limitations:**

yes

**Strengths And Weaknesses:**

**Strengths**:

1.**Rigorous Theoretical Foundation**: The paper provides a deep theoretical analysis of the root causes of instability (eigenvalue accumulation) and over-smoothing in dynamic graphs, accompanied by formal proofs demonstrating that the proposed damping mechanism guarantees convergence and stability.

2.**Simplicity and Efficiency**: The proposed solution is elegantly simple, achieving state-of-the-art performance by introducing a targeted control mechanism without adding significant computational complexity or architectural overhead.

**Weaknesses**:

1.**Suboptimal Presentation**: The paper’s organization obscures its core insight; by leading with the solution rather than the problem analysis, it risks misleading readers into viewing the method as a trivial addition of channel-wise decay factors, rather than recognizing the novelty of the symmetric/skew-symmetric operator decomposition.

2.**Insufficient Empirical Isolation**: While the theory argues for the necessity of each component, the paper lacks comprehensive ablation studies to empirically isolate their contributions. Without this, the method appears merely as a black-box nonlinear combination of graph kernels and self-evolution, making it difficult to discern which specific element drives the performance gains absent the theoretical context.

---

> ### Author Rebuttal · Authors · 2026-03-31
>
> We thank the reviewer for their positive evaluation of our work. We are encouraged by your recognition of our **rigorous theoretical analysis** and that our **elegant, simple, and efficient** solution **prevents signal explosion without sacrificing memory capacity**. We also appreciate your remark that our mechanism achieves **state-of-the-art performance on multiple benchmarks**. We welcome your constructive feedback, for which we respond to each concern point‑by‑point below.
>
> ### **Regarding W1**
> We thank the reviewer for their help in improving the presentation of the key contribution of our work. We have revised our paper (specifically, introduction and method sections) to reflect the suggestion, which we believe improves the clarity and impact of our claims.
>
> ### **Regarding W2**
> We thank the reviewer for the feedback. To empirically isolate components, we extended the ablation in Table 3 on ColoredLeafCounting, as shown below:
>
> | **Model** | **$T=200$** | **$T=400$** | **$T=600$** |
> |-|-|-|-|
> | **Three-colors** | | | |
> |$\text{LAMP}_{\text{adaptive}}$|$0.057_{ \pm 0.015 }$|$0.264_{ \pm 0.199 }$|$0.223_{ \pm 0.071 }$|
> |$\text{LAMP}_{\text{fixed}}$|$0.065_{ \pm 0.030 }$|$0.523_{ \pm 0.291 }$|$0.235_{ \pm 0.080 }$|
> | $\text{LAMP}_{\Gamma = 0}$ | $0.051_{ \pm 0.008 }$ | $0.520_{ \pm 0.145 }$ | $0.212_{ \pm 0.064 }$ |
> | $\text{LAMP}_{\text{NoWC}}$ | $1.327_{ \pm 0.020 }$ | $1.071_{ \pm 0.017 }$ | $0.650_{ \pm 0.257 }$ |
> | $\text{LAMP}_{\text{NoWC}, \beta=0}$ | $1.330_{ \pm 0.021 }$ | $1.053_{ \pm 0.026 }$ | $0.718_{ \pm 0.178 }$ |
> | **Binary-color** | | | |
> | $\text{LAMP}_{\text{adaptive}}$ | $0.467_{ \pm 0.083 }$ | $0.394_{ \pm 0.130 }$ | $0.831_{ \pm 0.136 }$ |
> | $\text{LAMP}_{\text{fixed}}$ | $0.421_{ \pm 0.063 }$ | $0.432_{ \pm 0.105 }$ | $1.135_{ \pm 0.239 }$ |
> | $\text{LAMP}_{\Gamma = 0}$ | $0.438_{ \pm 0.054 }$ | $0.438_{ \pm 0.091 }$ | $0.983_{ \pm 0.422 }$ |
> | $\text{LAMP}_{\text{NoWC}}$ | $0.848_{ \pm 0.017 }$ | $0.797_{ \pm 0.003 }$ | $1.373_{ \pm 0.000 }$ |
> | $\text{LAMP}_{\text{NoWC}, \beta=0}$ | $0.847_{ \pm 0.018 }$ | $0.793_{ \pm 0.009 }$ | $1.373_{ \pm 0.001 }$ |
>
> Specifically, we evaluate 5 LAMP variants, in which we incrementally relax our model: (i) the full version with adaptive dissipation, (ii) fixed dissipation, (iii) no dissipation ($\Gamma = 0$), (iv) no dissipation and no weight constraints (\*_NoWC), and (v) no dissipation, weight constraints and antisymmetric aggregation (\*_NoWC, $\beta=0$). The updated table clearly demonstrates the specific performance contribution of each theoretical element. Notably, the fully non-dissipative variant ($\Gamma = 0$) performs favorably in some tasks, as they inherently favor information retention over dissipation. However, the performance differences among the top configurations remain entirely within standard deviation margins.
>
> ### **Regarding Q1**
>  We thank the reviewer for this insightful connection to modern SSMs. We deliberately opted for a static, input-agnostic damping matrix $\Gamma$ to prioritize theoretical tractability and rigorously guarantee the dynamical system's stability. While input-dependent gating represents a promising avenue for handling instantaneous noise, our current design ensures that the model's baseline behavior remains precisely definable and analytically predictable.
>
> ### **Regarding Q2**
> We appreciate the suggestion to bridge our theoretical bounds with empirical stability through spectral analysis. Accordingly, we have added an eigenvalue spectrum plot of the trained model's Jacobian on the ChickenPox dataset in the revised paper (see https://anonymous.4open.science/r/LAMP-B4D1/eigenvalues_t0000.png). The plot displays all eigenvalues, visually confirming that they strictly reside in the stable left-half plane. Furthermore, the figure shows that the real parts of the eigenvalues, which govern stability and adaptive memory retention, are bounded below by $-0.5$. This lower bound directly correlates with the theory in Eq. (20) and the learned values of the diagonal damping matrix $\Gamma$ (0.6194 on average).
>
> ### **Regarding Q3**
> We thank the reviewer for this insightful question. As the reviewer correctly points out, higher-order graph shift operators (e.g., $A^k$) may exhibit unstable dynamics. This implies that not all choices of GSO are suitable: one must select operators whose powers neither diverge nor collapse to the null matrix (i.e., operators with spectral radius close to 1, such as the symmetrically normalized adjacency matrix). In this setting, the dissipative term $\Gamma$ can counteract the potentially unstable behavior induced by higher-order propagation. We believe that extending our framework to higher-order graph kernels is theoretically viable and represents a promising future direction to decouple instantaneous spatial propagation from the ODE solver depth.

---

> > ### Author Rebuttal · Reviewer_uuzf · 2026-04-03
> >
> > My concerns have been adequately addressed

---

### Official Review · Reviewer_fGbG · 2026-03-07

**Soundness:** 2
**Presentation:** 3
**Significance:** 3
**Originality:** 2
**Overall Recommendation:** 4
**Confidence:** 4

**Summary:**

The paper proposes LAMP, a differential-equation-inspired architecture for snapshot-based dynamic graphs that models latent evolution with an impulsive neural ODE: conservative antisymmetric flows between snapshots combined with a learnable, channel-wise dissipative term and a linear “impulse” update at observation times. The authors provide theory connecting the adaptive dissipation to contraction and input-to-state boundedness, give a sensitivity bound for spatio-temporal propagation, and present extensive experiments, including a new long-range synthetic benchmark and competitive real-world results on node forecasting and link prediction.

**Compliance With Llm Reviewing Policy:**

Affirmed.

**Key Questions For Authors:**

1. The paper states that the Jacobian of Eq. (2) has purely imaginary eigenvalues under the proposed parameterizations, but with nonlinear activations the Jacobian is D(h)Msk (minus Γ). Could you clarify precisely under what assumptions the “purely imaginary” statement holds, and otherwise revise to the contraction/log-norm view?
2. How do you choose ξ and μ in practice? Do you estimate ω during training to ensure ξ > e^{ωεL}? Please report the empirical values of e^{ωεL}/ξ over training and whether any schedule or adaptive rule is used.
3. How sensitive is performance to ε, L, and β? Are there stability issues or degradation for large L or small ε, and do you use any normalization or gradient clipping to control ω?
4. On real datasets with evolving topologies, beyond AS-733/Bitcoin-α, can you include a benchmark where both features and topology change (with node-level outputs) to isolate the benefits of the impulsive model?

**Limitations:**

Yes

**Strengths And Weaknesses:**

1.Introduces an impulsive ODE formulation for discrete-time dynamic graphs with a principled split between continuous conservative flows and discrete inputs. Combines antisymmetric parameterization with a learnable diagonal damping Γ, enabling adaptive, channel-wise control of memory retention vs. erasure.
2. Proposes a clear, stress-testing synthetic benchmark (ColoredLeafCounting) that jointly requires long spatial paths and temporal aggregation; LAMP achieves large, consistent gains.
Real-world evaluations span node forecasting and dynamic link prediction, with strong improvements in several cases and SOTA on link prediction.
3. The high-level architecture and the continuous/discrete interplay are well explained, and equations are easy to map to implementation.
4. The adaptive dissipativity idea could influence a broader class of DE-GNNs and temporal models that struggle with the stability–memory trade-off.

1. The global boundedness condition requires ξ > e^{ωεL}, where ω depends on learned parameters and the state via D(h). It is unclear how ξ is chosen a priori to guarantee the condition during training, and what guardrails ensure ω does not increase and violate the bound.
2. Comparisons omit several recent strong temporal baselines, e.g., state-space/SSM-based STGNNs (e.g., Mamba-style selective SSMs), recent continuous-time DE-GNNs for dynamic graphs (e.g., CTAN), and some Graph Transformers on snapshot forecasting, making it harder to contextualize gains. Runtime/memory comparisons are limited; forward-Euler with L steps per snapshot can be costly at large horizons, and more comprehensive profiling (vs. RNN/SSM/Transformer-style methods) would strengthen the case.
3. While the paper cites continuous-time DE-GNNs and stability literature, it does not empirically compare against recent non-attention long-range temporal models (e.g., selective state-space/Mamba-like STGNNs) or spiking/adaptive temporal models that emphasize stability and long-horizon propagation. Additional discussion would help position LAMP against continuous-time antisymmetric models (e.g., CTAN) and transition-graph approaches (e.g., TIP-GNN) in terms of modeling assumptions and applicability to snapshot vs. event streams.

---

> ### Author Rebuttal · Authors · 2026-03-31
>
> We thank the reviewer for the positive evaluation. We are encouraged by your recognition that **our principled ODE formulation and adaptive dissipativity could positively influence a broader class of DE-GNNs**, and that our stress-testing synthetic benchmark **effectively evaluates long spatio-temporal paths, alongside strong real-world results**.  We welcome your constructive feedback, for which we respond to each comment point‑by‑point below.
>
> ### **Regarding W1**
> We thank you for the comment regarding $\xi$ and Proposition 2.2. We restricted our search to $\xi \in $ {1, 2} to ensure system stability, avoiding $\xi < 1$ to prevent energy amplification. Practically, we did not explicitly constrain $\omega$ during training because, as detailed in our response to Question 2 below, empirical tracking of the learning dynamics demonstrates that the conservation holds without the need for adaptive scheduling or clipping.
> ### **Regarding W2 and W3**
> Thank you for the suggestion. While Table 5 already included Transformer baselines (STAEformer, STD-MAE) , following your comment we have expanded it with two SSM-based models (GGRNN, GraphSSM-S4) and an additional Transformer (PDFormer), as can be seen in the test MAE comparison in the Table below.
>
> ||PeMS-Bay|Metr-LA|
> |-|-|-|
> |STAEformer|1.88|3.34|
> |STD-MAE|1.77|3.40|
> |PDFormer|1.91|3.62|
> |GGRNN|2.34|3.88|
> |GraphSSM-S4|1.98|3.74|
> |**LAMP(ours)**|1.76|3.36|
>
> As shown in the table, LAMP strictly outperforms the three newly incorporated baselines on both datasets, while maintaining highly competitive performance against the best Transformers.
> Regarding computational profiling, efficiency is a core advantage (Section 2.3). Because time and memory scale linearly with discretization depth, we limit the forward Euler solver to $\leq 5$ steps, which empirically achieves peak performance while ensuring our complexity remains below Transformer architectures. A LAMP block with $L$ Euler steps matches the time complexity of an $L$-layer SSM or RNN.
> Again, following the reviewer’s suggestion we also expanded Related Work in the revised paper to include a broader discussion on continuous-time DE-GNNs (e.g. CTAN) and transition graph approaches (e.g. TIP-GNN).
> We have incorporated the above discussion, along with a new plot demonstrating our linear runtime scaling, into the revised version of the paper. Thank you.
> ### **Regarding Q1**
> We thank the reviewer for pointing out this ambiguity. To clarify, Eq. (2)'s Jacobian has purely imaginary eigenvalues only when the diagonal of $W_{sk}$ remains unaltered (as it is the product of a diagonal matrix and the antisymmetric $M_{sk}$). Our actual formulation (Eq. (7)) explicitly pushes these into the left-half plane. Following your question, we revised the paper to better refer to Appendix A.1, which shows the full derivation. Thank you.
> ### **Regarding Q2**
> Thank you for the insightful question. Tracking $e^{\omega\epsilon L}/\xi$ offers valuable insight into learning dynamics, and following your question we now plotted its evolution at the first and last epochs (see https://anonymous.4open.science/r/LAMP-B4D1/reg_analysis_cp_e1.pdf, https://anonymous.4open.science/r/LAMP-B4D1/reg_analysis_cp_e81.pdf), observing that while the Jacobian's logarithmic norm slightly increases (2.3 to 3.8), the conservation pattern holds. We added these results to the paper.
> In practice, we did not need to estimate $\omega$ during training or use adaptive scheduling to ensure stability. Furthermore, since $\mu$ regulates input magnitude, we fix $\mu = 1$ across tasks lacking random noise inputs.
> ### **Regarding Q3**
> We thank the reviewer. Accordingly, we conducted a grid search over $\epsilon$, $L$, and $\beta$ on TwitterTennis. Below, we isolate each parameter's effect on test MSE by fixing the others to their optimal values:
>
> |$\epsilon$|0.01|0.1|0.5|1.0|
> |-|-|-|-|-|
> |MSE|0.3003|0.2985|0.3415|0.3359|
>
> |$L$|1|2|4|8|16|
> |-|-|-|-|-|-|
> |MSE|0.3065|0.3033|0.3035|0.2985|0.2995|
>
> |$\beta$|-1.0|-0.1|0.0|0.1|1.0|
> |-|-|-|-|-|-|
> |MSE|0.2985|0.3036|0.3072|0.3026|0.3005|
>
> These results confirm that the step-size strictly bounds the error, degrading only at larger values, similar to classical numerical solvers for physical equations. For the full sweep, please look at https://anonymous.4open.science/r/LAMP-B4D1/tt_ebl_influence.pdf. Finally, we clarify that we do not use normalization or gradient clipping to control $\omega$.
> ### **Regarding Q4**
> Thank you for the question. We note that, beyond AS-733 and Bitcoin-$\alpha$, our evaluation includes TwitterTennis, a real-world benchmark featuring both evolving topologies and dynamic, non-synthetic features. We are not aware of any other real-world D-TDGs with these dual characteristics, making it the primary testbed to isolate our model's benefits.

---

> > ### Author Rebuttal · Reviewer_fGbG · 2026-04-01
> >
> > In response to the author's rebuttal, I maintain my original score.

---

### Official Review · Reviewer_bTkD · 2026-03-12

**Soundness:** 3
**Presentation:** 3
**Significance:** 3
**Originality:** 3
**Overall Recommendation:** 4
**Confidence:** 2

**Summary:**

This paper studies the problem of modeling dynamic graphs, where graph structure and properties could change over time.  The authors propose LAMP, a neural-ODE based model. The paper provides theoretical analysis of the proposed model, and empirical evaluation.

**Compliance With Llm Reviewing Policy:**

Affirmed.

**Key Questions For Authors:**

Please refer to strength and weakness.

**Limitations:**

I could not find the limitation section.

**Strengths And Weaknesses:**

### Soundness

- To me modeling dynamic graphs using neural-ODE is reasonable. How adaptive dissipativity module controls forgetting also makes sense to me.
- I did not check the proofs line-by-line. But at least theorem 2.1 and proposition 2.2 makes sense to me.


### Presentation

- The presentation of this paper is great.

### Significance

- I'm familiar with the problem that's being studied. Can the authors briefly discuss that in practice, which field or what type of tasks would be related?
- To me the empirical results demonstrate the significance of the proposed model.

---

> ### Author Rebuttal · Authors · 2026-03-31
>
> We thank the reviewer for their positive evaluation of our work. We are encouraged by your recognition that modeling dynamic graphs using a neural-ODE framework is **sound**, and that our adaptive dissipativity module provides a logical mechanism for controlling memory retention and forgetting. We also appreciate your remark that the paper's **presentation is great** and that the empirical **results demonstrate the significance of the proposed model**. We welcome your constructive feedback, that we address point-by-point below.
>
> ### **Regarding practical fields and tasks**
> We appreciate your insightful comment. In practice, discrete-time dynamic graphs are ubiquitous in domains where both the entities and their underlying topological relationships naturally evolve. We believe that the ability of LAMP to balance memory retention and noise filtering could be beneficial in multiple real-world applications such as:
> - **Transportation and Mobility:** Road networks where nodes (sensors/intersections) and edges (road segments) experience dynamically changing features like traffic volume or congestion. Accurate predictions require models to capture long-range temporal dependencies and spatial diffusion across the graph.
> - **Public Health:** Modeling the spread of infectious diseases over evolving human contact networks. Predicting future infection states relies heavily on tracking long-range spatial and temporal propagation as the network topology shifts.
> - **Agentic Systems and Graph Memories:** The development of autonomous AI agents operating in complex environments increasingly relies on dynamically evolving graph-structured knowledge and episodic graph memories [1,2]. In these settings, models require the ability to propagate information over long ranges to enable long-horizon reasoning. Simultaneously, as the agent's environment or objectives shift, the model must be able to selectively filter out or fade obsolete information. We believe that LAMP’s adaptive memory and dissipativity mechanisms may offer a principled approach to managing this precise balance.
>
> Following your comment, we have revised our paper and it now includes a discussion of these practical application domains of the studied problem and its limitations. Thank you.
>
> [1] Kurenkov et al. Modeling Dynamic Environments with Scene Graph Memory. In ICML 2023
>
> [2] Bei et al. Graphs Meet AI Agents: Taxonomy, Progress, and Future Opportunities. Preprint 2025

---

> > ### Author Rebuttal · Reviewer_bTkD · 2026-04-03
> >
> > concerns addressed by authors.

---

### Official Review · Reviewer_6QMw · 2026-03-13

**Soundness:** 4
**Presentation:** 4
**Significance:** 3
**Originality:** 3
**Overall Recommendation:** 5
**Confidence:** 4

**Summary:**

This paper studies directed graph modelling in dynamic settings. It focuses on discrete-time dynamic graphs, where the graph structure and the node or link features change over time. A central question in this area is how to extend graph learning methods from static graphs to dynamic graphs. The paper focuses in particular on information propagation over time, which is still less explored.

The authors propose LAMP, which stands for Long-range Adaptive Memory Propagation. LAMP is a new differential-equation-based graph neural network for dynamic graphs. Its model is based on a neural impulsive differential equation. It includes two parts: a neural vector field for spatial diffusion on the graph, and an impulse operator that updates the hidden state using newly observed features. In this way, the model balances memory retention and noise filtering.

The paper provides both empirical and theoretical results. Empirically, LAMP performs strongly on a new synthetic benchmark designed to test long-range propagation across both space and time, and it also does well on several real-world node-level and link-level prediction tasks. It matches or outperforms existing state-of-the-art methods for discrete-time dynamic graphs. Theoretically, the paper gives a sensitivity bound for LAMP, which supports effective propagation between nodes, and an asymptotic gain result that formalizes the trade-off between stability and memory capacity.

**Compliance With Llm Reviewing Policy:**

Affirmed.

**Key Questions For Authors:**

Can you please address the issue of computational performance (time / compute resources) of your approach vs. the ones you compare against?

**Limitations:**

None.

**Strengths And Weaknesses:**

The paper presents strong convincing work backed by both experiment and theory.

---

> ### Author Rebuttal · Authors · 2026-03-31
>
> We thank the reviewer for their thoughtful and positive evaluation of our work. We are encouraged by your recognition that **our approach is a novel way to balance memory retention and noise filtering for information propagation over time**, and that **our work is strongly backed by both comprehensive and theoretical results**. We also appreciate your remark that **LAMP successfully matches or outperforms existing state-of-the-art methods for discrete-time dynamic graphs**, which underscores the broad applicability and significance of our contribution. We welcome your constructive feedback, for which we respond to the concern below.
>
> ### **Regarding Computational Performance**
> We thank the reviewer for the insightful comment on computational performance: we agree that analyzing the trade-off between predictive accuracy and computational efficiency is crucial for evaluating dynamic graph models. We would like to direct the reviewer's attention to our complexity analysis in Section 2.3 and Appendix B.2, where we provide a detailed theoretical and empirical analysis of LAMP’s training and inference runtimes compared to the baselines. To summarize our empirical findings, we evaluated all models on a single NVIDIA V100 GPU with a fixed batch size of 64 to assess how runtime scales with increasing sequence lengths (T=200, 400, and 600) on the ColoredLeafCounting task. Our results demonstrate that LAMP offers a highly favorable balance of state-of-the-art accuracy and efficiency: LAMP scales linearly with sequence length. Its training and inference latencies are closely aligned with lighter models like A3T-GCN and T-GCN, and it is significantly faster than heavier recurrent baselines. For instance, at T=200, LAMP requires approximately one-third of the epoch training time compared to DCRNN (10.86s vs. 29.71s). Models like DynGESN, that exhibit slightly lower latencies due to their training-free message-passing layers, fail to capture the complex temporal dynamics required for the task, resulting in high error rates. Conversely, LAMP achieves a strong reduction in error (e.g., a Test MSE of 0.057 vs. the second-best DCRNN at 1.283 at T=200) without incurring the prohibitive computational costs typically associated with other top-performing models.
>
> If the reviewer feels this discussion is critical for the main narrative, we would be happy to move a condensed version of this performance analysis into the main text for the final camera-ready version, space permitting.

---

> > ### Author Rebuttal · Reviewer_6QMw · 2026-04-02
> >
> > Thank you for your answer.

---

### Decision · Program_Chairs · 2026-04-30

**Decision:**

Accept (spotlight)

**Comment:**

The paper proposes LAMP, a differential-equation-based architecture for dynamic graphs that models latent evolution with a neural ODE. Reviewers are enthusiastic about the paper. It presents an interesting approach to stability in dynamic graph learning.
Empirically, the method is well validated, including a thoughtfully designed synthetic benchmark (ColoredLeafCounting) and strong results across other tasks. The overall framework is clearly presented, and the adaptive dissipativity concept has the potential to influence a broad class of DE-GNN models.

Dynamical differential equations approaches on static graphs have been studied in the literature on [Energy Transformers](https://proceedings.neurips.cc/paper_files/paper/2023/hash/57a9b97477b67936298489e3c1417b0a-Abstract-Conference.html). Please contrast those methods with the proposed approach.

While it is possible to guess the meaning of DE-GNN acronym in the paper, it has never been defined - please spell it fully on first usage.